# Advanced sulfide solid electrolyte by core-shell structural design

Fan Wu [1], William Fitzhugh[1], Luhan Ye[1], Jiaxin Ning[1] & Xin Li[1]

Solid electrolyte is critical to next-generation solid-state lithium-ion batteries with high energy density and improved safety. Sulfide solid electrolytes show some unique properties, such as the high ionic conductivity and low mechanical stiffness. Here we show that the electrochemical stability window of sulfide electrolytes can be improved by controlling synthesis parameters and the consequent core-shell microstructural compositions. This results in a stability window of 0.7–3.1 V and quasi-stability window of up to 5 V for Li-Si-P-S sulfide electrolytes with high Si composition in the shell, a window much larger than the previously predicted one of 1.7–2.1 V. Theoretical and computational work explains this improved voltage window in terms of volume constriction, which resists the decomposition accompanying expansion of the solid electrolyte. It is shown that in the limiting case of a core-shell morphology that imposes a constant volume constraint on the electrolyte, the stability window can be further opened up. Advanced strategies to design the next-generation sulfide solid electrolytes are also discussed based on our understanding.

---

[1] John A. Paulson School of Engineering and Applied Sciences, Harvard University, Cambridge, MA 02138, USA. These authors contributed equally: Fan Wu, William Fitzhugh. Correspondence and requests for materials should be addressed to X.L. (email: lixin@seas.harvard.edu)

The fast development of portable electronic devices and electric vehicles demands lithium-ion batteries with high power, energy density, and safety[1–5]. Solid-state batteries using solid electrolytes such as polymers[6–8], oxides[9–12], or sulfides[13–16] are hence promising for next-generation lithium-ion batteries. The application requires solid electrolytes with good chemical compatibility, high Li-ion conductivity and a wide electrochemical stability window. High lithium-ion conductivity has been achieved in various solid electrolytes, including sulfide glasses, sulfide glass-ceramics, and crystalline sulfides. Sulfide glass solid electrolytes[17,18] were reported to have a lithium-ion conductivity of $0.1−1 \, \text{mS cm}^{-1}$. Sulfide glass-ceramics were produced by precipitation of crystalline phases from the precursor sulfide glasses to reduce the grain-boundary resistance. For example, the $Li_2S$-$P_2S_5$ glass-ceramic system with $Li_7PS_6$[19,20], $Li_7P_3S_{11}$[21], and $Li_{3.25}P_{0.95}S_4$[14] precipitates showed improved conductivities of over $1 \, \text{mS cm}^{-1}$. Crystalline sulfide solid electrolytes of $Li_{3.25}Ge_{0.25}P_{0.75}S_4$[22], $Li_{10}GeP_2S_{12}$ (LGPS)[16] and $Li_{9.54}Si_{1.74}P_{1.44}S_{11.7}Cl_{0.3}$ (LSPS-Cl)[13] were reported with the high conductivity of 2.2, 12, and $25 \, \text{mS cm}^{-1}$, respectively.

Despite the superior lithium-ion conductivity of LGPS and LSPS[13,16,23,24], various groups[23–26] reported the narrower stability windows of around 1.7–2.1 V, while others reported wider voltage windows[13,16,27]. We suggest that changes in the microstructure of the electrolyte materials or in the volume constriction condition of the battery cells may result in different voltage stability windows. Although the structure of the crystalline LGPS or LSPS has been studied by diffraction techniques[16,28], change in the synthesis details from various groups may decorate the same crystalline phase with different microstructures of amorphous or glassy phases. This is very possible considering that sulfide systems with mixed glass-ceramic phases can exist from synthesis[19–21] and there lacks a systematic check and understanding about the effect of such amorphous/glassy phases on the materials properties in the previous reports. An understanding of how these possible changes in microstructure can control the electrochemical properties of crystalline sulfide electrolytes, especially the voltage stability and interface compatibility, is hence critical. The goal of our paper is to demonstrate that the control and modification of the microstructures in LGPS and LSPS can adjust and improve their voltage stabilities. More importantly, we aim to reveal the underlying mechanism between the microstructure and the performance of sulfide solid electrolytes, which can serve as the guidelines for the future materials and battery cell designs.

Here, we show experimentally that the voltage stability of our core-shell LSPS sulfide solid electrolytes can be largely improved by modifying the composition of the amorphous shell that encloses the crystalline LSPS core. The shell compositions are controlled by adjusting the synthesis parameters. The shells with high silicon compositions increase the voltage stability window of the compounds, while low silicon compositions in the shells decrease the voltage stability. Using density functional theory (DFT) simulations, we further demonstrate that the major underlying mechanism of this phenomenon is that in the sulfide solid electrolytes with appropriate core-shell microstructures to provide the volume constriction on the materials level, the work necessary to accommodate the large local strains during decomposition exceeds the energy release of decomposition and, hence, the decomposition is not thermodynamically favorable, leading to enlarged voltage stability windows. The results herein also provide a design strategy from a predictive formalism to further stabilize sulfide solid electrolytes by microstructure modifications, and more generally by volume constriction or pressurization that can be realized on both materials and battery cell levels.

## Results

**Theoretical rationale**. Our computational results reveal that the sulfide electrolytes tend to expand during decay, leading to a positive "reaction strain," which is defined by Eq. (1) in terms of the volume of the decay products ($V_d$) and of the initial solid electrolyte ($V_{SE}$). In some cases, the reaction strain for LSPS is predicted to reach the levels as high as 56%.

$$\epsilon_{RXN} = \frac{V_d - V_{SE}}{V_{SE}} > 0 \,. \tag{1}$$

In the case of a rigid shell, because the decomposed products are larger than the pristine material and the shell forbids total particle expansion, any partial decomposition must compress the remaining pristine material enough as to make room for the decomposed products. If the decomposition energy (energy above the hull) is less than the work needed to adequately compress the surroundings, then this reaction is energetically forbidden. The effective compressibility of the shell defined by Eq. (2) represents a metric for the performance of the shell in terms of volume constriction. The limit of $\beta_{shell} \to 0$ represents a rigid shell allowing zero volume expansion, whereas $\beta_{shell} \to \infty$ recovers the no shell condition.

$$\beta_{shell} = \frac{1}{V_{core}} \frac{\partial V_{core}}{\partial p} \,. \tag{2}$$

Note that $\beta_{shell}$ is therefore not the shell's material compressibility, but an effective compressibility of the core-shell structure. It is a function of not only the material properties but also the geometry of the shell, including the curvature and thickness. We show that a low effective compressibility provided by the core-shell structure will suppress solid electrolyte decomposition with large enough reaction strain, a mechanism that can effectively widen the voltage stability window. An amorphous shell with high Si composition falls into this category based on the high Young's modulus of amorphous Si reported[29,30] previously.

At the onset of decay, where the fraction of the decomposed material is approximately linear with pressure (SI equation (11)), Eqs. (1) and (2) imply that there is a direct mapping from the fraction of decomposed product to the internal pressure of the core-shell system as well as the volume. Therefore, for the core-shell morphology with low compressibility, expansion of the particle into any neighboring region or void requires significant strain energy. Thermodynamically comparing the energies of the decomposed, expanded, void-filling state with that of the pristine solid electrolyte state, it is the latter that is more energetically favorable. The core-shell structure is thus stable at zero pressure, with no tendency to expand.

**Characterization of core-shell microstructure**. Cl-doped LSPS powders ($Li_{9.54}Si_{1.74}P_{1.44}S_{11.7}Cl_{0.3}$, or LSPS-Cl) were synthesized at seven different annealing temperatures ranging from 400 to 500 °C. The as-synthesized LSPS-Cl materials were confirmed to have the same LGPS-type crystal structure[16] (space group P4$_2$/nmc, 137) by X-ray diffraction (XRD) (Fig. 1). Comparable full-width at half-maximum of the Bragg peaks suggest close particle sizes among the LSPS-Cl powder samples, which is consistent with the scanning electron microscopy (SEM) images (Fig. 2a) showing similar morphology and particle sizes.

Both SEM and transmission electron microscopy (TEM) (Fig. 2a, b) images of LSPS-Cl annealed at 450, 460, 480, and 500 °C (hereafter LSPS-Cl 450, 460, 480, and 500) show the particles with the core-shell structure. Nano-particles embedded inside the amorphous shells (Supplementary Figure 1) are shown to be single-crystalline by high-resolution TEM (HRTEM) images

(Supplementary Figure 2). To measure the composition of different regions (i.e. core, shell and particle in the shell), scanning TEM (STEM) energy-dispersive X-ray spectroscopy (EDS) analyses were performed on multiple areas inside different particles for each material (Supplementary Figure 3). The analyzed atomic compositions of LSPS-Cl 450, 460, 480, and 500 obtained from STEM EDS are summarized in Fig. 2c and Supplementary Figure 4, showing that all regions of these particles are comprised of Si, P, S, and Cl (note that lithium cannot be detected by EDS technique). There are relatively small variations in the elemental compositions of the core regions among the four samples, compared with the obvious compositional difference in the shells. As the annealing temperature increases from 450 to 500 °C, the atomic concentration of silicon in the shell generally decreases from ~40 to <10%, while that of sulfur increases from ~40 to ~80%.

**Electrochemical stability and battery performance.** Cyclic voltammetry (CV) was used to experimentally evaluate the electrochemical stability of LSPS-Cl materials in low-voltage (0.1–2.0 V) and high-voltage (1.0–5.0 V) ranges. Figure 3a shows that in the low-voltage range, all seven LSPS-Cl materials show a peak at ~0.7 V with similar current densities, indicating the reactions or decompositions at almost the same voltage with similar intensity. However, the high-voltage-range CV tests (Fig. 3b) show obvious difference among the seven LSPS-Cl materials. The current densities of LSPS-Cl 480, 490, and 500 increase dramatically beyond ~3.1 V, indicating severe decompositions of these materials at high voltages, while the LSPS-Cl 450, 460, and 470 materials show little current densities at high voltages. Figure 3c magnifies Fig. 3b by around 500 times, demonstrating the high-voltage decompositions beyond 3.1 V of LSPS-Cl 450, 460, and 470 samples are at least two orders of magnitudes smaller than other samples. The onset voltage of the decomposition peak is

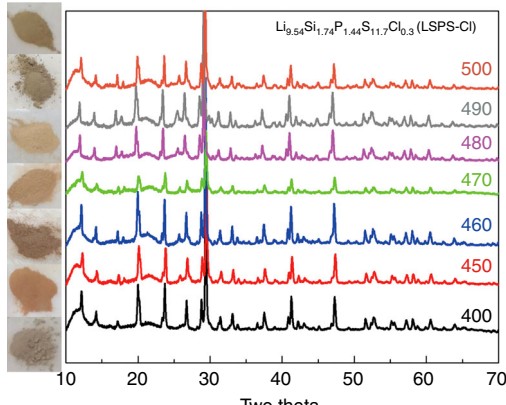

**Fig. 1** Color under visible light and XRD patterns of the LSPS-Cl powders. The LSPS-Cl powder changes color along with the annealing temperature from 400 to 500 °C. All the XRD patterns can be indexed by the same space group, P4$_2$/nmc (137)

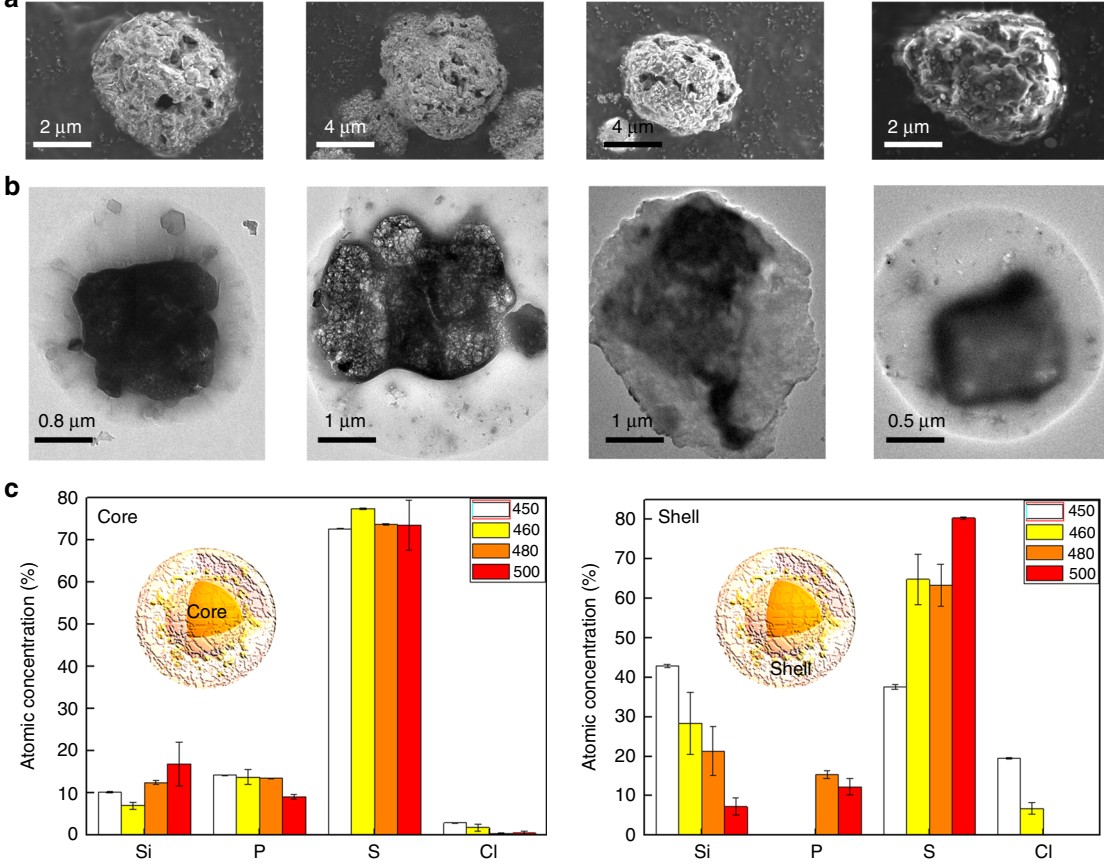

**Fig. 2** Microstructural study on the morphology and composition of LSPS-Cl powders annealed at 450, 460, 480, and 500 °C. **a** SEM and **b** TEM images show their typical core-shell structures. **c** STEM EDS analyses of these samples show a continuously decreasing Si concentration and increasing S concentration in the shell of these samples versus the increasing annealing temperature

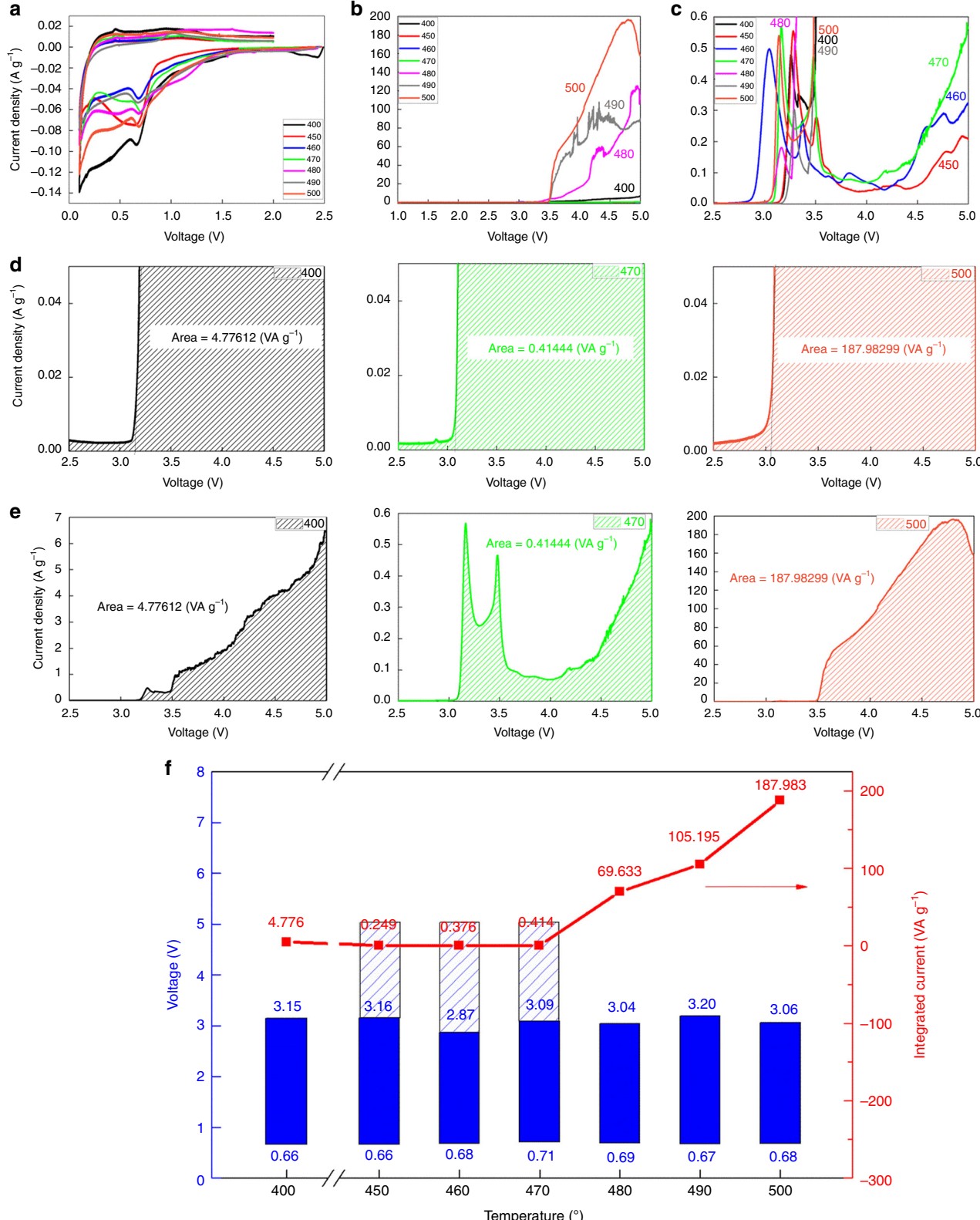

**Fig. 3** Cyclic voltammetry (CV) test and analyses of the LSPS-Cl samples annealed at various temperatures. **a** Low-voltage (0.1−2.0 V) and **b**, **c** high-voltage (1.0−5.0 V) stabilities of LSPS-Cl materials. **d** Initial decomposition onset voltage and **e** integral spectrum intensity of three representative LSPS-Cl materials (LSPS-CL 400, 470, 500) in the categories of minor, medium, and severe decompositions. **f** Summary diagram to show the core-shell-structured LSPS-Cl materials with similar voltage stability window of ~0.7–3.1 V. The decomposition of minor-decomposition materials (LSPS-Cl 450–470) above 3.1 V is largely suppressed, giving a quasi-stability window up to 5 V

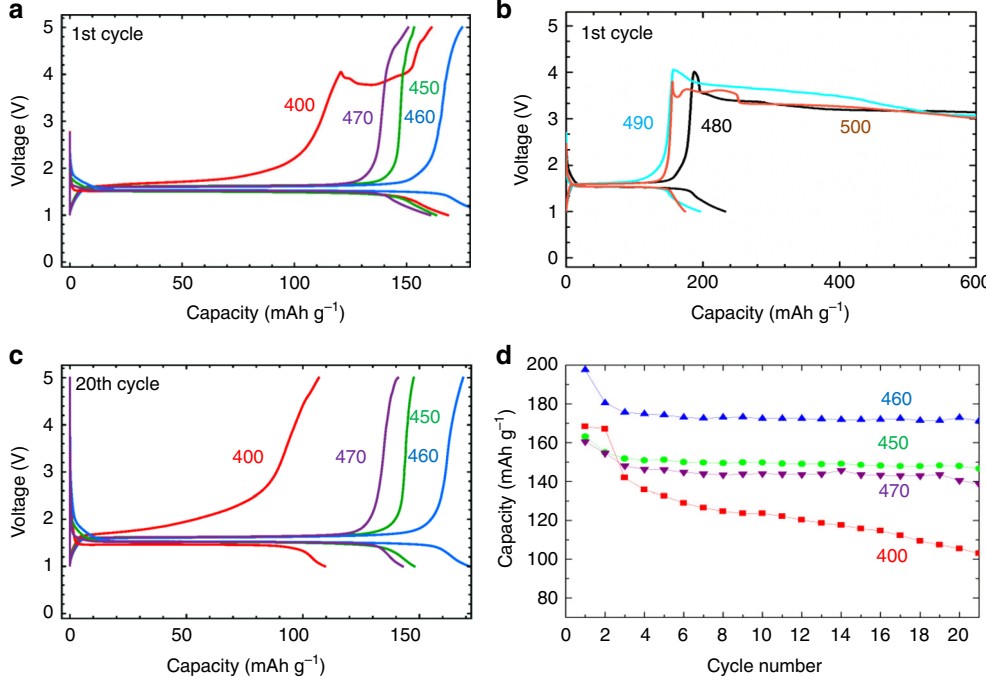

**Fig. 4** Battery performance of $Li_4Ti_5O_{12}$+LSPS-Cl+C/LSPS-Cl/glass fiber/Li cells incorporating different LSPS-Cl materials annealed at different temperatures. **a** First and 20th-cycle charge/discharge curves for batteries incorporating medium- and minor-decomposition materials (LSPS 400,450–470). **b** First-cycle charge/discharge curves for batteries incorporating severe-decomposition materials (LSPS 480–500). **c** 20th-cycle charge/discharge curves for batteries incorporating (LSPS 400,450–470). **d** Cycle performance of the batteries incorporating LSPS 400,450, 460, and 470 within the first 20 cycles

defined by the cross section of the two tangent lines drawn from the horizontal baseline and the current density curves near the onset for each LSPS-Cl material (Fig. 3d and Supplementary Figure 5). They all show the onset voltages near 3.1 V, with a maximum value for LSPS-Cl 490 at 3.20 V and a minimum value for LSPS-Cl 460 at 2.87 V. Therefore, the voltage stability windows of these LSPS-Cl materials with core-shell structure are all from 0.7 to around 3.1 V as summarized in Fig. 3f, much larger than the 1.7~2.1 V window reported previously[23–26].

As discussed in Fig. 3b, c, despite the similar stability voltage windows for the seven LSPS-Cl materials, their decompositions above 3.1 V are completely different. To quantify the difference, the seven current density−voltage curves were integrated from 2.5 to 5 V to obtain the integrated current (Fig. 3e, f, Supplementary Figure 6 and Supplementary Table 1), according to which the seven LSPS-Cl materials can be classified into three categories: minor-, medium-, and severe-decomposition. LSPS-Cl 480, 490, and 500 are severely decomposed samples with the integrated currents in the range of $70-188$ VA g$^{-1}$. In comparison, the materials (LSPS-Cl 450, 460, and 470) with minor decompositions show the integrated currents that are several hundred times smaller, ranging from 0.25 to 0.41 VA g$^{-1}$. This indicates that the decomposition of LSPS-Cl 450, 460, and 470 above 3.1 V is successfully suppressed, leading to a quasi-stable voltage window up to 5 V. LSPS-Cl 400 is the only sample in the medium-decomposition category with an integrated current of 4.8 VA g$^{-1}$.

The different stabilities of LSPC-Cl are also reflected in the battery cycling test between 1 and 5 V incorporating the seven different LSPS-Cl materials (see Methods). Figure 4a shows the first-cycle voltage curves of LSPS-Cl 400, 450, 460, and 470 batteries with all of them being able to charge up to 5 V. The LSPS-Cl 450, 470 batteries can cycle between 1 and 5 V

smoothly, consistent with their high-voltage stabilities up to 5 V. While an obvious voltage bump appears at 4 V on the first charge profile of LSPS-Cl 400 battery, most probably due to the medium-degree decomposition of LSPS-Cl 400 at high voltages. In contrast, due to the severe decomposition of LSPS-Cl 480, 490, and 500, the corresponding batteries cannot be charged above 4 V and failed eventually within the first cycle. Only the batteries incorporating minor/medium-decomposition materials (LSPS-Cl 400, 450, 460, and 470) can cycle smoothly between 1 and 5 V for multiple cycles. Figure 4c shows the voltage profiles of the 20th cycle for these batteries, in which the 1.5 V charge plateau remains flat for LSPS-Cl 450, 460, 470 batteries, while that of LSPS-Cl 400 battery deforms significantly. The cycle performance of these batteries (Fig. 4d) also shows that the specific capacity of LSPS-Cl 400 battery decays faster than that of LSPS-Cl 450, 460, 470 batteries, consistent with their voltage stabilities analyzed earlier. The derivative of capacity versus voltage (dQ/dV) plots for the seven batteries (Supplementary Figure 9a) show peaks at 1.5 V for all of them (due to the phase transition of LTO), and peaks at 3.5–4 V only for batteries incorporating medium- or severe-decomposition materials, corresponding well to the CV results (Fig. 3b). dQ/dV of these batteries cycled between 0.1 and 2 V (Supplementary Figure 9b) further confirms the ~0.7 V decomposition peak for all the seven LSPS-Cl materials, again consistent with the CV results (Fig. 3a). Note that the LSPS-Cl samples with high-voltage stabilities coincide with the high lithium-ion conductivities measured by impedance spectroscopy. LSPS 400, 460, and 480 were selected as the representative materials of each category for the ionic conductivity measurement. Results (Supplementary Figure 10) show that LSPS 460 has the highest ionic conductivity of 3.1 mS cm$^{-1}$, while LSPS 400 and 480 show relatively lower ionic conductivity of 2.28 and 2.39 mS cm$^{-1}$, respectively. Note that even higher ionic conductivity may be

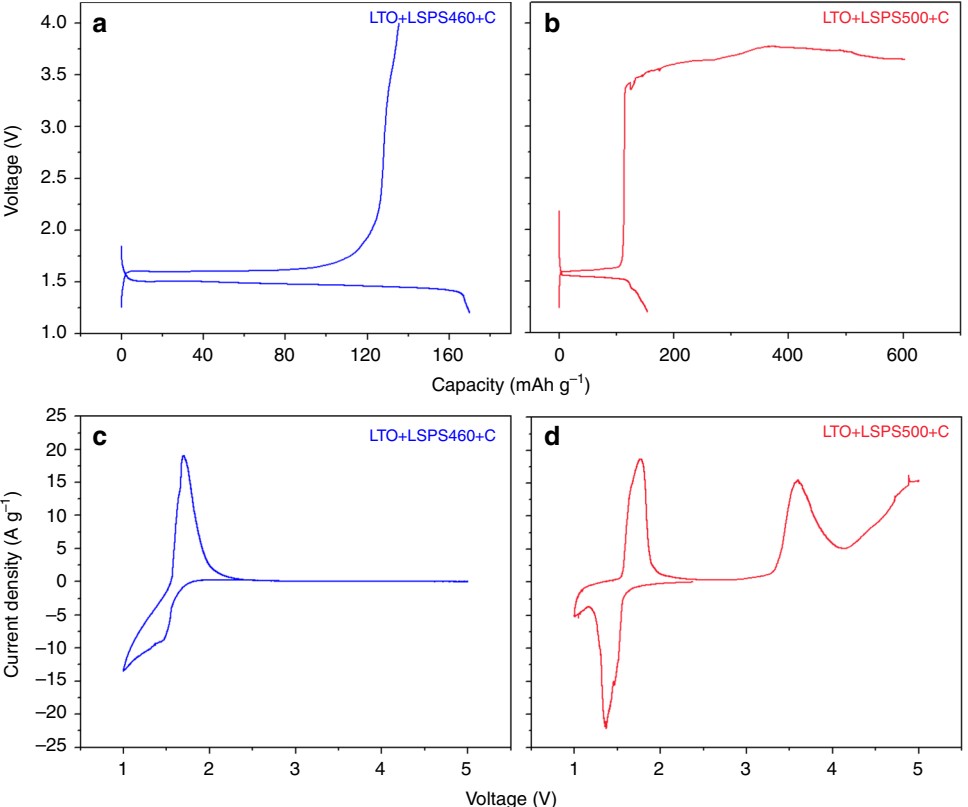

**Fig. 5** Battery performance and CV test results of all-solid-state batteries using: **a**, **c** LTO+LSPS460+C as cathode, LGPS as solid electrolytes and LPS-coated lithium foil as anode. **b**, **d** The same tests for LSPS500. Current rate for battery test is 0.1 C. Sweeping rate for CV test is 0.1 mV s⁻¹

obtained by applying higher pressure[27] during the impedance measurement, which was not applied during our test (see Methods).

It is worth noting that for the above-mentioned CV and charge/discharge tests, the half-cell batteries contain a glassfiber separator to avoid the influence from the interfacial interaction between the LSPS and lithium metal. Small amount of liquid electrolyte is thus added to the glassfiber to allow the lithium-ion conduction. However, to rule out any effect of the liquid electrolyte that may permeate into the cathode layer, all-solid-state battery without any glassfiber separator layer was fabricated and tested. Special treatment was performed on Li foil to form a protective layer (see Methods) so that the interfacial reaction between LSPS solid electrolyte and Li can be avoided. Results show that all-solid-state battery using a mixture of LTO +LSPS460+C powers as cathode (Fig. 5a) can be charged to 4 V smoothly, while the all-solid-state battery using LTO+LSPS500 +C cathode failed at ~3.6 V (Fig. 5b) due to a much more severe decomposition of LSPS500. These all-solid-state battery results correspond very well to the trend obtained by semi-solid-state battery configuration (Fig. 4), confirming again that LSPS solid electrolytes synthesized with different conditions do have different electrochemical stabilities.

Making use of the same solid-state battery configuration, solid-state CV test was performed to rule out the possible effect of liquid electrolyte on the electrochemical stability test results of LSPS. Figure 5c, d shows the CV scan results in the voltage range of 1–5 V for all-solid-state batteries using LTO+LSPS460+C and LTO+LSPS500+C cathode, respectively. The CV peaks of LTO around 1.5 V exist for both batteries. However, no decomposition can be observed for LSPS460, while the decomposition of

LSPS500 starting at ~3.5 V is very strong and obvious. These results match well with the CV results obtained by semi-solid cell, confirming the different electrochemical stabilities of various LSPS solid electrolytes. More importantly, the semi-solid-state battery configuration is proved to be a valuable configuration in terms of evaluating material properties, such as the voltage stability of solid electrolytes. To show the oxidation/decomposition products of LSPS solid electrolytes, and that the minor-decomposition LSPS does have true larger electrochemical stability, more evidence is demonstrated in Supplementary Figures 11–14.

**Theoretical simulation results**. The ability of a core-shell structure to stabilize the sulfide solid electrolytes via mechanical constriction was predicted due to the large reaction strains defined in Eq. (1). For example, at 0 V vs. lithium metal, LSPS decays or decomposes to $Li_3P$, $Li_{21}Si_5$, $Li_2S$ with a reaction strain of 56.2%. Conversely, at 3.5 V vs. lithium metal, LSPS decays to $P_2S_5$, $SiS_2$, $S$ with a reaction strain of 21.5%. Similar reaction strains are found for LGPS as well as doped derivatives of both LGPS and LSPS. If conditions are created such that the work necessary to accommodate this strain is greater than the decay energy, then the decay cannot proceed and the stability window will be expanded.

One route, in theory, to increase such work and prevent the decay is to apply a constant pressure. If the solid electrolyte is pressurized, then any volume expansion must exert certain mechanical work. Such a constant pressure ensemble can be directly calculated via DFT simulations. The relevant free energy, at zero temperature, for a lithium open system at constant

pressure is:

$$\Phi(p, \mu_{Li}) = H(p) - \mu_{Li} N_{Li}, \quad (3)$$

where $H$ is the enthalpy of the structure (calculated via DFT) and $N_{Li}$ is the number of lithium atoms in the cell. Results (Supplementary Figures 15–16) clearly show that applying large pressures can improve the stability window of sulfide solid electrolytes considerably.

Alternatively to this direct-pressure approach, consider the limit of a perfectly rigid shell enclosing a particle of solid electrolyte. The constraint of the shell on the core in this extreme is that of constant volume. If a portion of the particle decays, and hence expands, an internal pressure within the core must be created that compresses the neighboring solid electrolyte (SE) sufficiently as to make the room for the volume expansion associated with the reaction strain. For the decay of a particle with a perfectly rigid shell, the decomposition energy is given in terms of the fraction of pristine SE ($x_{SE}$) and decomposed SE ($x_d$) by the following equation:

$$\Delta\phi_V(\mu) = x_{SE}\,\phi_{SE}(p, \mu) + x_d\,\phi_d(p, \mu) - \phi_{SE}(0, \mu). \quad (4)$$

Given that the total volume cannot change, the pressure must depend on the fraction of decomposed SE such that:

$$x_d V_d(p) + x_{SE} V_{SE}(p) = V_{SE}(p = 0) = V_{core}. \quad (5)$$

In the limit of $x_d \ll 1$ (e.g. at the onset of decay), the pressure dependence of the decay fraction can be shown to be $x_d(p) = p\beta_{SE}\epsilon_{RXN}^{-1}$, where $\beta_{SE}$ and $\epsilon_{RXN}$ are the compressibility and decomposition reaction strain of the solid electrolyte, respectively (detailed derivations are in Supplementary information). When combined with Eq. (4), this leads to an expression of the volume constrained decay energy, $\Delta\phi_V$, in terms of the constant pressure decay, $\Delta\phi(p, \mu)$, at the onset of decay (Eq. 6). Ultimately, this allows evaluation of the volume restricted decay energy, which cannot be calculated directly from DFT.

$$\Delta\phi_V(\mu) \approx \frac{p\beta_{SE}}{\epsilon_{RXN}}\left(\Delta\phi(p, \mu)\right) + \left(\phi_{SE}(p, \mu) - \phi_{SE}(0, \mu)\right). \quad (6)$$

For this decay to be energetically favorable (e.g. for Eq. (6) to be negative), the constant pressure decay energy must exceed the work done to compress the remaining solid electrolyte.

$$|\Delta\phi(p, \mu)| > \frac{\epsilon_{RXN}}{p\beta_{SE}}\left(\phi_{SE}(p, \mu) - \phi_{SE}(0, \mu)\right). \quad (7)$$

The conclusion of this inequality is that in order for solid electrolyte to decompose inside of a perfectly rigid shell, the decomposition energy must be sufficiently high as to be able to compress the remaining solid electrolyte enough to accommodate the larger volume of the decomposed products.

To further generalize this conclusion derived for a perfectly rigid shell to an elastic shell, the decay energy is modified to be

$$|\Delta\phi(p, \mu)| > \frac{\epsilon_{RXN}}{p(\beta_{SE} + \beta_{shell})}\left(\phi_{SE}(p, \mu) - \phi_{SE}(0, \mu)\right) \quad (8)$$

in which $\beta_{shell}$ is the effective compressibility of the shell as defined in Eq. (2). The decay conditions for both isobaric (no shell) and isovolumetric (ideal shell) can be recovered by taking the shell compressibility to infinity or zero, respectively. Finally, Eqs. (7) and (8) can be reduced in the limit of $p = 0 \to \delta p$ (i.e. at

the onset of decay). Defining the effective bulk modulus $K_{eff} = (\beta_{SE} + \beta_{shell})^{-1}$ and noting that $V = \partial_p\phi(p, \mu)$, the inequality of Eqs. (7, 8) becomes a familiar stress−strain type relation:

$$\frac{|\Delta\phi(p, \mu)|}{V} > K_{eff}\,\epsilon_{RXN}. \quad (9)$$

Our DFT simulations, in conjunction with grand canonical post-processing and further analysis based on the above formalism (see Methods), show that while LSPS decay remains largely unchanged with doping and initial composition in a zero-pressure isobaric environment, which shows a narrow voltage stability window similar to LGPS at zero pressure[25,26] (Fig. 6a), the application of a shell with certain rigidity in our case can greatly improve the stability window (Fig. 6b). Additionally, Eq. (1) may in fact underestimate the reaction strain as the decay converts a single-crystal solid electrolyte into a polycrystalline mixture. In the latter case, perfect packing is unlikely. If the decay products would have a packing efficiency of $\eta$, then the reaction strain would be given by Eq. (10). Figure 6c illustrates the significant potential impact of such a packing efficiency to further open up the voltage stability window, suggesting the additional importance of different decay processes within the rigid shell picture.

$$\epsilon_{RXN} = \frac{V_d\eta^{-1} - V_{SE}}{V_{SE}}. \quad (10)$$

## Discussion

Experiment and computation have been shown here to agree that while solid electrolytes are plagued by narrow stability windows, microstructured materials, namely core-shell structures in this particular case, show significantly improved performance. These results suggest a new direction for solid-state Li battery development and that for such core-shell microstructures, three overarching conditions are needed to significantly increase the stability of solid electrolytes.

1. The effective compressibility of the shell must be low. The ideal shell structure would be completely rigid ($\beta_{shell} = 0$), which forces any decomposition to exert work on the neighboring solid electrolytes, leading to an increased threshold for the decomposition energy.
2. The compressibility of the pristine sulfide must be low. By making the pristine solid electrolyte more rigid, it requires more work to be locally compressed.
3. Reaction strains should be maximized. $\epsilon_{RXN}$ indicates how much of the neighboring solid electrolytes must be compressed for decomposition to occur, multiplying the effects of condition 2.

Conditions 2 and 3 are intrinsic to the solid electrolytes, whereas condition 1 is dependent on the microstructure and composition of the shell. Supplementary Figure 17 shows biaxial modulus data for all Li-Si-P-S materials available within the Materials Project database[31]. The biaxial moduli were obtained via the KVRH approximation. The overall trends show that, within this family of compounds, materials tend to increase in modulus with increasing Si content. Conversely, these materials tend to become lower in modulus (higher in compressibility) as either the S or P content is increased.

These results suggest that excess Si would be beneficial to the stability of sulfide solid electrolyte. Given that Si and P sit in the same site, this excess Si would have to come at the expense of P. In fact, this is in line with the best known performing $Li_{10}SiP_2S_{12}$ derivative, $Li_{9.54}Si_{1.74}P_{1.44}S_{11.7}Cl_{0.3}$, which was successfully cycled

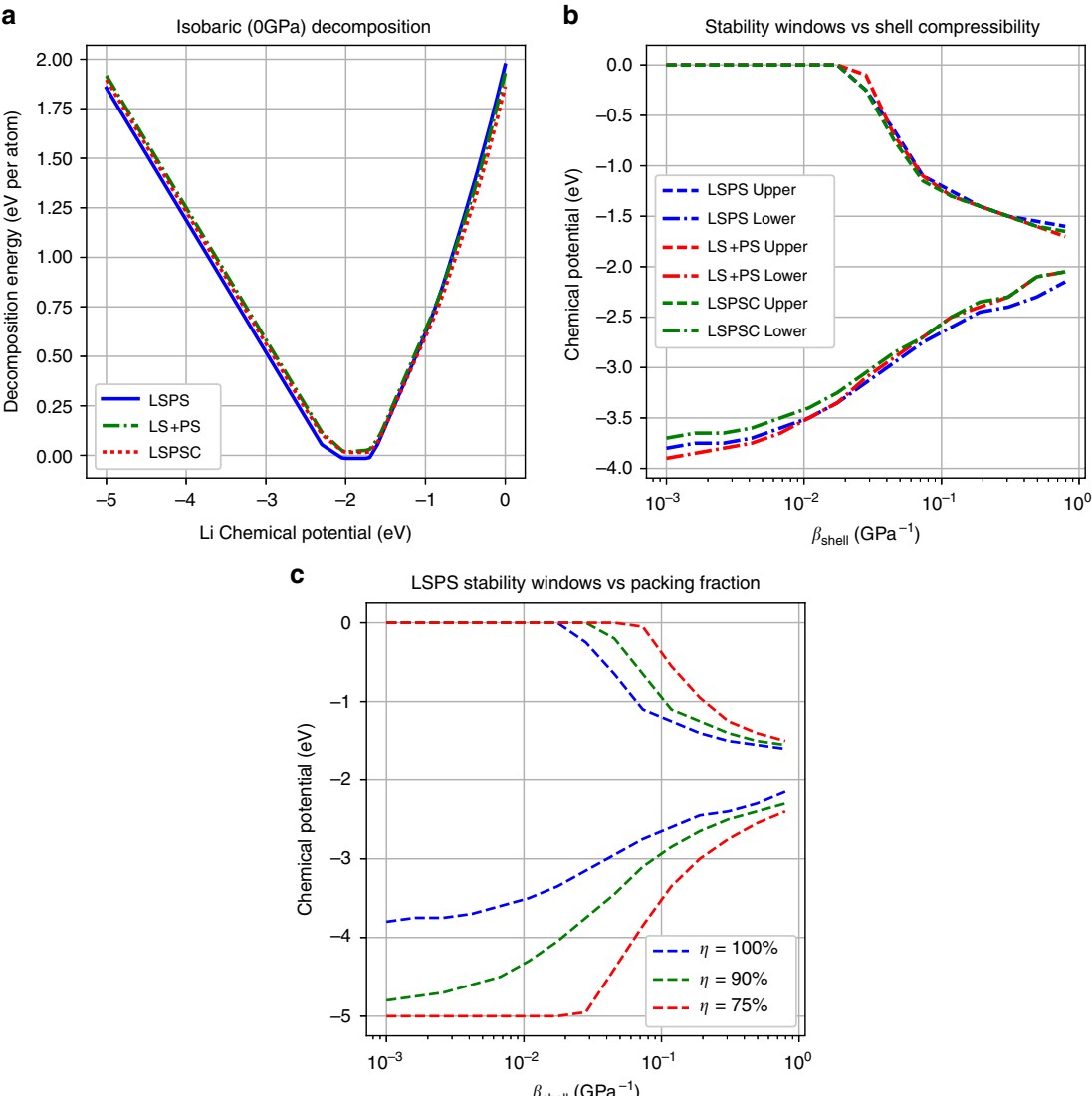

**Fig. 6** Computational simulation results showing the effects of doping and shell compressibility on stability windows of LSPS-based solid electrolytes. **a** Zero-pressure decomposition energy for $Li_{10}SiP_2S_{12}$ (LSPS), $Li_{10}Si_{1.5}P_{1.5}S_{11.5}Cl_{0.5}$ (LSPSC), and $Li_{10}Si_{1.5}P_{1.5}S_{12}$ (LS+PS). **b** Stability windows for LSPS, LSPSC, and LS+PS as a function of shell effective compressibility. The zero limit of $\beta_{shell}$ (left extreme of x-axis) corresponds to a perfectly rigid shell (isovolumetric decay). $\beta_{shell} \gg \beta_{core}$ (right extreme of x-axis) corresponds to no shell (isobaric decay). **c** Stability window for LSPS at different decay packing fractions. Stability was only considered within the range of 0–5 V vs. lithium

over 500 times[13]. Note that the Si:P ratio is over twice here than a typical LSPS.

It is also interesting to note that if we simply consider the monovalent defects $\left(V_{Li}^{1-}, Si_P^{1-}, P_S^{1+} Cl_S^{1+}\right)$, the general doping structure is $Li_{10-x}Si_{1-x+y}P_{2+x-y}S_{12-y}Cl_y$. For the doping structure to have $[V_{Li}] = 0.46$, $[Cl_S] = 0.3$, and for everything to remain in its ideal valence state, the Si, P, and S concentration would be fixed at $Li_{9.54}Si_{0.84}P_{2.16}S_{11.7}Cl_{0.3}$. This composition likely has an increased rigidity compared to $Li_{10}SiP_2S_{12}$ given the excess Si. Forcing even more Si (such as the compound we synthesized) could lead to an amorphous shell phase with Si rich, consistent with our STEM-EDS measurement.

Given the high modulus of amorphous Si (biaxial modulus ~140 GPa[29,30]), it is likely that such Si distribution in the core-shell LSPS would lead to both a core of decreased compressibility and a relatively rigid shell. A spherical core-shell model with the morphology and composition consistent with our TEM observation corresponds to a $\beta_{shell}$ value of around $3 \times 10^{-2}$ GPa$^{-1}$,

giving an estimated voltage stability window of 0.7–3.1 V based on Fig. 6b. The estimation here used a shell modulus of around 75 GPa, which is obtained based on an unusually high Si composition of around 40 at% as characterized by our STEM-EDX (Fig. 2) and using the modulus vs. Si composition relationship in Supplementary Figure 17. In comparison the Si composition in the core of our crystalline LSPS is only less than 10 at%. It should also be noted that even if the shell modulus is comparable to the core (20–30 GPa), the stability window of a core-shell-structured solid electrolyte is still significantly larger than that without shell at an estimated $\beta_{shell}$ value of around $10^{-1}$ GPa$^{-1}$ based on Fig. 6b.

The decreased compressibility of the core and a rigid shell here in our core-shell LSPS thus would satisfy both conditions 1 and 2 for improved stability windows of sulfide solid electrolyte. Lastly, given that Si has a larger atomic radius than P, it would not be surprising that Si-rich LSPS would have an increased reaction strain. To confirm the latter, we compared the zero pressure

isobaric decompositions for $Li_{10}SiP_2S_{12}$ and $Li_{10}Si_{1.5}P_{1.5}S_{12}$. The result was that in the low/high chemical potential region, the Si-rich material had a 2.3–3.6% higher reaction strains than the non-Si-rich material.

Our simulation of the LGPS system (Supplementary information) show that its voltage stability can also be improved by the same mechanism, suggesting the universality of the theory for the sulfide solid electrolytes. Note that it is also possible that the "passivation layer" mentioned in the previous literatures[25,32] actually serves as a similar core-shell microstructure generated by the reaction on the surface of solid electrolyte. However, we want to emphasize that our mechanism of solid electrolyte stabilization is beyond the conventional passivation layer argument. In order to be useful in a lithium conductive system, any passivation layer must itself be lithium conductive. Given that the lithium stability window of bulk solid electrolyte is determined by the lithium chemical potential, the adoption of a lithium potential conserving passivation layer may have minimal impact on the chemical tendency towards decay. A quantitative description of how a passivation layer could directly lead to improved stability remains undemonstrated. However, if the passivation layer is made to be rigid, even if only comparable to the core (20–30 GPa), recovering the core-shell structure discussed herein, then the passivation layer could both be lithium conductive and inhibit lithium reactions via mechanical, rather than chemical, stabilization.

It is worth noting that in a constant pressure system, the strain energy is simply proportional to the volume expansion. However, in a quasi-constant volume system as represented by our core-shell structure before any actual decomposition, an infinitesimal change in decomposition fraction ($\delta x_d$) requires an infinitesimal increase in strain energy ($\delta U_{strain}$), given by both an increase in pressure and volume. Within the voltage stability window predicted by Fig. 6, the relation of $\delta U_{strain} \geq |\delta \Delta G_{reaction}|$ is always satisfied, giving no actual decomposition, internal pressure nor particle swelling at 0 GPa external pressure.

In closing we wish to highlight that what we have shown here is that the pressurization accompanying decomposition within a core-shell morphology can require an amount of work that cannot be provided by chemical decomposition within the expanded voltage stability range. It is also entirely possible that there are additional energetic factors that result from this mechanism, which could explain our high voltage stability vs. lithium. For example, lithium conductivity and lithium configurational entropy are directly related[33]. Given that the lattice parameter constriction is known to drastically reduce conductivity[26], it is possible that pressurization due to decomposition leads to lithium ordering in the nondecomposed fraction of the particle. This lithium ordering would of course oppose the reaction with a magnitude given by the change in lithium configurational entropy. Similar to how the work associated with pressurization inhibits any decomposition in which the work is greater than the chemical reaction energy, the addition of entropic effects would inhibit any decomposition when the decay energy is not sufficient to overcome both the work and the entropy decreases. This is also possibly extended to other terms that are dependent on pressure (i.e., defects, surface interactions, kinetics).

Practically, we expect a macroscopic sulfide solid electrolyte material to contain particles with a distribution of $\beta_{shell}$. It should be noted that the shell serves as a pressure vessel for the sulfide solid electrolyte core. As such, the effective compressibility of shell is a complicated function of the shell mechanical properties, thickness, curvature, core volume, etc. In general, however, shell structures with high moduli, such as amorphous silicon, are preferred for enhanced voltage stabilities. Considering the different lithium-ion conductivities in the microstructures, if the shell conductivity is lower than the core, a thin rigid shell is thus preferred, in addition to the criteria for the core of low compressibility and high reaction strain, for the design of next-generation sulfide solid electrolytes. As a final note, we wish to highlight that while this work has focused on microstructure volume constraints, other methods could be used to take advantage of this theoretical understanding. For example, many works have shown that sulfide electrolytes show strong performance when used in battery cells that are designed for the application of pressure.[34–36]

## Methods

**Synthesis**. The starting materials used for the synthesis of $Li_{9.54}Si_{1.74}P_{1.44}S_{11.7}Cl_{0.3}$ solid electrolyte were $Li_2S$ (>99.9% purity, Alfa Aesar), $P_2S_5$ (>99% purity, Sigma Aldrich), $SiS_2$ (>99% purity, American Elements), and $LiCl$ (>99% purity, Alfa Aesar). All of the reagents were weighed in the appropriate molar ratio and then placed into a $ZrO_2$ ball-mill jar containing $ZrO_2$ milling balls All of the procedures were conducted under an argon atmosphere inside a glove box. The mixture was then mechanically milled using a planetary ball milling facility for 40 h. Following the ball milling procedure, the mixture was sealed into glass tubes and then heated at 400, 450, 460, 470, 480, 490, and 500 °C, respectively, for 8 h, followed by a slow cooling procedure back to room temperature.

**Electrochemistry characterization**. The cyclic voltammograms (CV) of Li/Glassfiber/LSPS-Cl+C/Au cells were measured between 0.1 and 2 V for low-voltage range and 1–5 V for high-voltage range at a scan rate of 0.1 mV s$^{-1}$ on a Solartron electrochemical potentiostat (1470E), using lithium as reference electrode. The obtained current densities of CV tests for different LSPS-Cl were obtained by normalizing to 1 g of LSPS-Cl with the same area[2]. For battery performance test, the composite cathode was prepared by mixing $Li_4T_5O_{12}$, LSPS-Cl, Polytetrafluoroethylene (PTFE) and carbon black with a weight ratio of 60:30:5:5. This mixture was then rolled into a thin film. LSPS-Cl thin film was prepared by mixing LSPS-Cl and PTFE with a weight ratio of 95:5. The Swagelok battery cell of cathode film/LSPS-Cl film/glass fiber/Li was then assembled in an argon-filled glove box. The piece of glass fiber separator was inserted between LSPS-Cl film and Li metal foil to avoid interfacial reaction between them in order to best reflect the intrinsic stability difference among various core-shell LSPS-Cl materials. One drop of 1 M $LiPF_6$ in ethylene carbonate (EC) and dimethyl carbonate (DMC) solution (1: 1) was carefully applied onto the glass fiber to allow the lithium-ion conduction through the separator. The galvanostatic battery cycling test was performed on an ArbinBT2000 work station at room temperature. The specific capacity was calculated based on the amount of LTO (60 wt%) in the cathode film. For impedance measurement, C-LSPS/LSPS/C-LSPS cell was fabricated by sandwiching and cold pressing (applied force 0.6 ton, sample cross area = 0.316 cm$^2$) an electrolyte powder layer with two carbon black-electrolyte powder layers. 50 wt% carbon black was mixed with electrolyte to form the conductive layer, with the purpose of decreasing the interface resistance. The measurement was performed in a Solartron electrochemical potentiostat (1470E + 1455 FRA). The frequency range of the measurement was from 100 Hz to 1 MHz and the amplitude was 10 mV. For all-solid-state batteries, lithium anode was coated with a protection layer in the following steps: Firstly, a solution was prepared by mixing $Li_2S_6$ (0.5 M) and $P_2S_5$ (1 M) in the mixture solvent of DME and DOL (1:1, v/v) in the glove box for 1 h. Then lithium foil (5/16 inch in diameter) was soaked into the prepared solution for 2 h. The soaked lithium foils were assembled into Li/Li symmetric battery for five cycles of charge-discharge, with 1 M LiTFSi, 0.025 M $Li_2S_6$, and 0.05 M $P_2S_5$ in DME and DOL (1:1, v/v). Finally, the lithium foils after the above process were taken out of the symmetric battery for use as the anode (hereafter LPS-Li). The cathode of the all-solid-state cell is a mixture of LTO + LSPS + carbon black powder (75:20:5), and solid electrolyte is pure LGPS powder. 0.5 mg of cathode mixture powder and 80 mg of solid electrolyte powder were pressed together to form a disc-shape pellet, which is pressed onto a freshly prepared LPS-Li in a Swagelock Cell to form an all-solid-state battery. The same battery configuration was used for solid-state CV test with 0.1 mV s$^{-1}$ sweeping rate.

**(Micro)structural analysis**. XRD data were obtained using a Rigaku Miniflex 6G with a Cu target X-ray source (wavelength = 1.54056 Å). The various LSPS-Cl powders were placed onto standard XRD sample holders and sealed with Kapton film and vacuum grease under an argon atmosphere in a glove box. Structural parameters were refined in the Topas software by using Rietveld refinement technique. TEM samples were prepared by dropping the as-synthesized powder directly to the TEM copper grid, sealed inside the airtight bottles in the glove box and opened immediately before loading into the TEM column with an air exposure less than 30 s. The Gatan vacuum transfer TEM sample holder was used to double check, which confirms the same results. JOEL 2010F was used for TEM and STEM

EDS characterization on multiple particles for each LSPS-Cl sample, and the average composition values of the obtained data were statistically analyzed.

**Density functional theory calculations**. In order to allow comparability with the Material Project crystal database, all DFT calculations were performed using the Material Project criteria[31]. All calculations were performed in VASP using the recommended Projector Augmented Wave (PAW) pseudopotentials. An energy cutoff of 520 eV with k-point mesh of 1000 per atom was used. Compressibility values were found by discretely evaluating the average compressibility of the material between 0 and 1 GPa. Enthalpies were calculated at various pressures by applying external stresses to the stress tensor during relaxation and self-consistent field calculations.

**Pre/post-process calculations**. The Python Materials Genomics (pymatgen) library was used for pre/post-processing of high-throughput calculations[37]. In particular, the phase diagram modules were used to calculate the convex hull for each elemental system. All possible oxidation (lithium extraction) and reduction (lithium insertion) reactions were accounted for by modifying the free energy as governed by Eq. (3)[38]. For isobaric calculations (Supplementary Figures 15 & 16), $H(p)$ is calculated at every pressure for both LGPS and decomposition products. Then the convex hull is determined using Eq. (3). In isovolumetric calculations, we are interested in the onset of decay, which is equivalent to there being zero radial pressure ($\lim x_d \rightarrow 0, p = 0$). Pymatgen was used in conjunction with the Materials Project API[39] for accessing the Materials Project crystal database's Materials Explorer[40]. Biaxial moduli were calculated from the Poisson's ratio and $K_{VRH}$ approximation of the bulk modulus.

## Data availability

The datasets generated during the current study are included in this published article (and its supplementary information files) or are available from the corresponding author on reasonable request. The datasets analyzed, but not generated, in the current study are available from the Materials Project, MaterialsProject.org.

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

## Acknowledgements

The authors thank Zhigang Suo and Joost J. Vlassak for helpful discussions. This work was supported by the Dean's Competitive Fund for Promising Scholarship at Harvard University. The SEM and TEM experiments were conducted at Center for Nanoscale Systems (CNS) at Harvard University and Center for Materials Science and Engineering (CMSE) at MIT, supported by the National Science Foundation. The computation was supported by the Extreme Science and Engineering Discovery Environment (XSEDE) and the Odyssey cluster supported by the FAS Division of Science, Research Computing Group at Harvard University.

## Author contributions

F.W., L.Y., and J.N. performed experiments. W.F. conducted theoretical calculations. X.L. supervised the research. All authors discussed the results and wrote the manuscript.

## Additional information

**Competing interests:** The authors declare no competing interests.

