## [Peer Review file · Nature Communications]

Reviewers' comments:

Reviewer #1 (Remarks to the Author):

The authors report on a core shell structure design of sulphide solid electrolytes to achieve increased electrochemical stability based on the internal pressure. In general the relation between the electrochemical potential and the internal pressure is known, but hard to make use of because huge pressures required. However, the concept to apply this to the electrochemical stability is interesting. My main concern is that in my opinion the experiments reported do not prove the hypothesis, and I have fundamental concerns that it will work in realistic solid state batteries:

(1) The electrochemical tests to prove the larger electrochemical stability are performed with a liquid electrolyte. As such the electrochemical activity as measured by the CV scans will strongly depend on the electrochemical surface area and porosity. From the images in Figure 2a, it is not unlikely that these differ substantially between the different material, however, this is not reported nor discussed.

(2) With the internal pressures predicted, why would the material not expand? The ability to expand will bring back the normal, small electrochemical stability window. Even in solid state batteries one should expect still sufficient voids to expand, which makes it doubtful that this concept would work in practice.

(3) It is puzzling that the solid electrolytes are not tested in realistic solid state battery geometries, as this is where one would like to see the advantage. To proof the concept, solid state batteries should have been prepared and electrochemically cycled, preferably with higher voltage cathodes.

(4) Above 3 V, but most likely already below, part of the Li will be extracted (the solid electrolyte will be oxidised), which should be expected to influence the calculated pressures and stability window significantly, however this is not considered.

(5) In the "Characterization of core-shell microstructure" section, what is the exact composition of the core and shell materials. It appears that the core-shell structures are obtained by direct mechanical milling process. Typically for these type of materials the pure phase is obtained by the high rotation speed milling process. These 'pure' phase is a core-shell structure, where the core is typically more crystalline and the shell more amorphous. This may also have a strong influence on the apparent stability, which may be of large relevance for this work. More analysis of the core shell structure therefore appears necessary.

(6) Recent literature has shown the electrochemical activity of sulphide electrolytes, which may explain why the discharge capacity of solid-state battery using the LSPS-Cl 460 electrolyte delivers a larger discharge capacity than the theoretic capacity of LTO (165mAh/g) in Figure 4b. It is therefore vital that the materials are analysed after charge, and after cycling to proof true larger electrochemical stability.

Based on the fundamental issues raised I conclude that, although the concept sounds interesting, I do not consider it proven by the reported experiments and much more experiments need to be conducted.

Reviewer #2 (Remarks to the Author):

The authors have performed a combined experimental and theoretical study exploring the possibility

of using compression from a 'shell' on a sulphide Li-ion solid-electrolyte 'core' to suppress reductive decomposition of the solid-electrolyte, thereby increasing its electrochemical stability window.

I think the idea proposed is very interesting. As such I think it is warranted for publication in Nature Materials.

I have a few comments in general, noted below, as well as concerns about the computations presented in this manuscript, especially the description of the computational work, which I feel needs to be addressed before the paper is accepted:

1. It is not clear if the plot shown in SI Fig. 11 & 12 is obtained by computing $H(p)$ (Eq. 3) for just LGPS OR for LGPS AND all the decomposition products? This is important to know so that one can ascertain the claims made in the paper.
2. There is also not much details about how the constant pressure DFT calculations were performed. Citing Ref. 25 is not adequate to reproduce these results. In fact, I suggest the authors explicitly report the $H(p)$ vs p graphs for all the phases considered.
3. The mobility of Li-ions should also depend on the pressure, and this will in turn influence the conductivity, in addition to the stability. In fact pressure should reduce mobility. Can the authors incorporate possible change in kinetics to clearly discuss the pros- and cons- of using such a core-shell structure to improve chemical stability?
4. Much of the known superionic Li-ion solid-electrolytes have intrinsic site-occupancy disorder, which is known to influence the high mobility seen in these materials (e.g. J. Mater. Chem. A, 2017,5, 1153-1159). Can the authors comment on how this site occupancy could possibly be affected by the induced pressure from the shell ? (this is somewhat related to comment-3 above).

Reviewer #3 (Remarks to the Author):

This paper describes the importance of amorphous phase in LGPS superionic conducting solid electrolyte. The properties of sulfide solid electrolytes can be controlled and improved by modifying the microstructures. The amorphous shell covering the LSPS core improves the stability window of the composite solid electrolyte particles. It seems that description of originality is excessive. Glass ceramic solid electrolytes, which contain high ion conducting crystals in the glass matrix, have been studied the most. The authors should state progress and new ideas for prior research with proper references. $\text{Li}_2\text{S-P}_2\text{S}_5$ glass-ceramic solid electrolytes with LGPS phase has already been reported around 2002 although the crystal structure of the LGPS phase has been clarified since 2011. Since there are many other concerns, I do not recommend publication in the present stage.

- The major claims are unclear in this manuscript.
- Some references should be added and discussed.
- It seems that description of the originality is excessive. An intelligible explanation is important, but originality should be described accurately.
- There are many unclear points in experiments and results.
- The evidence of decomposition in oxidation should be shown.
- What is "glass fiber" inserted between Li and solid electrolyte? If the glass fiber includes liquid electrolytes, the impregnation with electrolyte is most concerned.
- Since this paper deals with concrete discussions, interest in researchers except battery community is

considered to be relatively small.

-Details on DFT calculations are not provided.

General response to reviewers:

A reoccurring concern expressed by the reviewers is regarding the feasibility of operating a solid electrolyte (SE) at high pressures (e.g., “reduced conductivity”, “expansion into voids”, “hard to make use of because huge pressures required” etc). While each case of concern will be commented on below, we wish to clarify that in the case of our core-shell structured SE, the decomposition can be stabilized in the absence of any pressure. Our core-shell structure prevents the decomposition tendency of SE, which makes our SE stable with zero pressure in practice. As you will see in the following detailed point-to-point response to reviewers, we show that although we have no external pressure added in our all-solid-state battery, our LSPS SE can still transport lithium ions (Figure 1.5) through the solid electrolyte layer and within the cathode layer when charged to high voltages.

It is critical to understanding our theory to consider a spectrum of shell possibilities that range from no shell (e.g. a constant pressure system determined by the experimental environment) to a perfectly rigid shell (e.g. a constant volume system determined by the coating). In reality, every shell exists in a condition between constant pressure and constant volume, so we generalize to a quasi-constant volume shell that transitions from a constant pressure to constant volume based on the metric ‘effective compressibility’ (eq 2).

In the constant pressure, no-shell, extreme (revised Supplementary Table 2 and Supplementary Figure 15) stabilization requires tremendous experimentally applied pressure. In this case, the stabilization would occur by experimentally applying energy to the material in the form of mechanical work. We agree that this method is not practically useful. However, the calculations do suggest the possibility of utilizing large reaction strains to stabilize the material.

The second extreme, that of a rigid shell, maintains much more promise as it does not pressurize at all within the given stability window. In this case, because the decomposed products are larger

than the pristine material, and the shell forbids total particle expansion, the decomposition must compress the remaining pristine material enough as to make room for the decomposed products. If the decomposition energy (energy over the hull) is less than the work needed to adequately compress the surroundings, then this reaction is energetically forbidden. It is precisely the range of the lithium chemical potential such that the decomposition energy is less than the required work that defines the stability window. Hence, within the stability window, decomposing is energetically forbidden. Accordingly, the particle will not pressurize within this window – it is only once this window is exceeded that the decomposition/pressurization will occur.

It is the latter of these two extremes that we have aimed to recognize experimentally.

Reviewers' comments:

Reviewer #1 (Remarks to the Author):

The authors report on a core shell structure design of sulphide solid electrolytes to achieve increased electrochemical stability based on the internal pressure. In general the relation between the electrochemical potential and the internal pressure is known, but hard to make use of because huge pressures required. However, the concept to apply this to the electrochemical stability is interesting. My main concern is that in my opinion the experiments reported do not prove the hypothesis, and I have fundamental concerns that it will work in realistic solid state batteries

As mentioned above in our general response to reviewers, while pressurization would lead to stabilization, core-shell morphologies can be stabilized without the realization of any pressure. The experiments discussed below realize this theoretical prediction, with vastly improved lithium stability in the absence of any applied pressure. Also, our sulfide SE has been successfully applied into realistic solid-state battery with zero external pressure. Details can be found in our response to the following questions.

(1): The electrochemical tests to prove the larger electrochemical stability are performed with a liquid electrolyte. As such the electrochemical activity as measured by the CV scans will strongly depend on the electrochemical surface area and porosity. From the images in Figure 2a, it is not unlikely that these differ substantially between the different material, however, this is not reported nor discussed.

We understand the reviewer's concern, however, we will show that the electrochemical stability of LSPS we discussed in the manuscript is not an artifact of the possible coexisting small amount of liquid electrolyte by the following three experiments.

I. Prior to a battery test, the battery was held at 3V for 24 hours to decompose all the specially-designed liquid electrolyte within the high voltage portion (i.e. cathode) of the battery. The specially-designed liquid electrolyte is 1M LiPF_6 in EC:DEC:HMPA(1:1:1), as the additive of HMPA greatly reduces the overall voltage stability of the liquid electrolyte [1-2] The decomposition of the specially-designed liquid electrolyte can be seen in the comparison of galvanostatic battery cycling tests (Figure 1.1), where the battery lost the capacity in the normal liquid electrolyte cell without any LSPS solid electrolyte, while it can cycle in the cell with

coexisting solid electrolyte (LSPS450, stable at high voltages including 3V in our original manuscript) and (decomposed) liquid electrolyte. If LSPS450 is replaced by LSPS500 (solid electrolyte with severe decomposition at high voltages but still stable at 3V in our original manuscript), the battery can still cycle. Moreover, XRD test results of LSPS450 and LSPS500 thin films after 3V 24h holding (Figure 1.2) confirm that they remain stable after the potentialstatic holding. Therefore, this is a clear evidence that in the cathode region the solid electrolyte (such as LSPS450 and LSPS500) is the only lithium conductive medium after the liquid electrolyte decomposition.

Figure 1.1: Charge/discharge profiles of batteries using either (a) LTO+C+PTFE or (b) LTO+LSPS450+C+PTFE or (c) LTO+LSPS500+C+PTFE as cathode and Lithium foil as anode. One piece of glassfiber separator was added between cathode and anode with 1 drop of liquid electrolyte (1M LiPF₆ in EC/DEC/HMPA=1:1:1) added to conduct Li through the separator. All batteries were held at 3V for 24h before cycling between 1 and 2V at 0.5C rate.

Figure 1.2: XRD patterns of (a) LSPS450/500+C+PTFE thin film before, (b) LSPS450+C+PTFE thin film after and (c) LSPS500+C+PTFE thin film after potentialstatic holding at 3V for 24h in a battery using LSPS+C+PTFE thin film as cathode, Lithium foil as anode and one piece of glassfiber as separator.

Meanwhile, after holding at 3V for 24 hours, the subsequent CV scan will also reflect the intrinsic materials property of LSPS without being influenced by the liquid electrolytes. The results of the new CV test (with 24-hours holding at 3V before CV scan) for minor-decomposition material (LSPS450) and severe-decomposition material (LSPS500) are shown here in Figure 1.3 for comparison. The integrated decomposition current densities for LSPS450 and LSPS500 are 0.055805 VA/g and 932.971786 VA/g, respectively, consistent with the trend reported in our original manuscript. Different electrochemical surface area and porosity won't be able to explain the huge difference in the measured decomposition current density after all the liquid electrolyte in the high voltage region has been pre-decomposed. We thus exclude the influence by the electrochemical surface area and conclude that the reported differences in the voltage window are the intrinsic materials property of different LSPS.

Figure 1.3: New CV results for minor-decomposition material (LSPS450) and severe-decomposition material (LSPS500). Before these CV tests between 1 and 5V, the batteries were held at 3V for 24h to completely decompose the liquid electrolytes possibly existing in pores and on surface area of LSPS particles. Therefore the new CV results show the true electrochemical stability of LSPS solid electrolytes, without being influenced by different amounts of liquid electrolytes trapped in different surface areas of various LSPS materials. The ramping speed of new CV test is 0.1mV/s, same with that reported in original manuscript.

II. We also performed the potentialstatic holding test for batteries using commercial liquid electrolyte (1M LiPF₆ in EC/DEC=1:1). As the commercial electrolytes have wider voltage stability window [3-5] than the special liquid electrolyte we prepared above, and decomposition only happens at voltages above 4.5V [4,5], we chose to hold the battery at 6V for 10 hours to decompose all the commercial liquid electrolyte within the high voltage portion of the battery prior to a battery cycling test. Note that based on our original argument and theoretical prediction, we expect neither LSPS450 nor LSPS500 solid electrolyte to be structurally completely stable at the 6V holding, however, LSPS450 should be much more stable than LSPS500.

Similar to the 3V holding case, the battery lost the capacity in the normal liquid electrolyte cell without any LSPS solid electrolyte (Figure 1.4a). But it can cycle well in the cell with coexisting solid electrolyte (LSPS460, argued as more stable at high voltage in our original manuscript) and (decomposed) liquid electrolyte (Figure 1.4b). If LSPS460 is replaced by LSPS500 (solid

electrolyte with severe decomposition at high voltage in our original manuscript), the battery lost the capacity again after the 6V holding (Figure 1.4c). XRD test results show that both LSPS460 and LSPS500 show certain decomposition after the potential static holding at 6V for 10h, which is understandable and consistent with our prediction in the original manuscript. However, the results here support our argument in the original manuscript that LSPS460 is a comparably much more stable solid electrolyte than LSPS500 at high voltage.

Figure 1.4: Charge/discharge profiles of batteries using either (a) LTO+C+PTFE or (b) LTO+LSPS460+C+PTFE or (c) LTO+LSPS500+C+PTFE as cathode and Lithium foil as anode. One piece of glassfiber separator was added between cathode and anode with one drop of electrolyte (1M LiPF₆ in EC/DEC 1:1) added to conduct Li through the separator. All batteries were held at 6V for 10h before cycling between 1 and 2V at 0.5C rate.

Note that in the above-mentioned battery configuration for our CV test and potential static holding battery performance test, only a small amount of liquid electrolyte was directly dropped to the separator to make it lithium conductive to the lithium metal anode, while the cathode layer containing LSPS was not directly added with any liquid electrolyte. Although small amount of liquid electrolyte may be absorbed from the separator to the cathode film, the potential static holding will decompose all the liquid electrolyte there, making it a new type of solid state battery, as the conventional liquid electrolyte battery won't be able to pass the potential static holding test like what is shown in Figure 1.1 and 1.4.

III. Moreover, as you will see in our response to your comment#3, pure solid-state CV test (Figure 1.6) was performed to rule out the possible effect of liquid electrolyte on the electrochemical stability test results of LSPS. In our all-solid-state battery configuration, neither liquid electrolyte nor glass fiber is added. Results show intense decomposition of LSPS500 starting at ~3.5V, but no decomposition for LSPS460, which matches with the CV results obtained by liquid cell, confirming once more the different electrochemical stabilities of various LSPS solid electrolytes.

- [1] Suzette Izquierdo-Gonzales, Wentao Li, Brett L. Lucht, Hexamethylphosphoramide as a flame retarding additive for lithium-ion battery electrolytes, *Journal of Power Sources*, **135**, 291–296, (2004).
- [2] K. Xu, Electrolytes and Interphases in Li-Ion Batteries and Beyond, *Chem. Rev.* **114**, 11503–11618, (2014)
- [3] J.R. Croy, A. Abouimrane, Z. Zhang, Next-generation lithium-ion batteries: The promise of near-term advancements, *MRS Bull.* **39**, 407-415, (2014).
- [4] Y. Li, B. Ravdel, B.L. Lucht, Electrochem. Electrolyte Reactions with the Surface of High Voltage LiNi_{0.5}Mn_{1.5}O₄ Cathodes for Lithium-Ion Batteries, *Solid State Lett.* **13** A95-A97, (2010).

[5] L. Hu, Z. Zhang, K. Amine, Electrochemical investigation of carbonate-based electrolytes for high voltage lithium-ion cells, *J. Power Sources*, **236**, 175-180 (2013).

(2) With the internal pressures predicted, why would the material not expand? The ability to expand will bring back the normal, small electrochemical stability window. Even in solid state batteries one should expect still sufficient voids to expand, which makes it doubtful that this concept would work in practice.

We have calculated the thermodynamics of solid electrolyte (SE) decay from an unstrained SE state ($p = 0$) to a strained (compressed, $p > 0$) SE + decomposed product state.

$$SE(p = 0) \leftrightarrow x_{SE}SE(p > 0) + x_d d(p > 0)$$

It is important to note that the interior pressure of the particle, the fraction of decomposed product and the volume of the particle are all dependent. At the onset of decay or decomposition ($\lim x_d \rightarrow 0$):

$$x_d V_d(p) + x_{SE} V_{SE}(p) = V_{core}(p) \approx V_{core}^0 + \beta_{shell} V_{core}^0 p \quad \#(SI. 1)$$

$$p \approx \frac{x_d \epsilon_{RXN}}{(\beta_{shell} + \beta_{SE})} \quad \#(SI. 11)$$

Therefore, while the SE particle could expand into the surrounding void, it could only do this by decaying ($x_d = 0 \rightarrow \delta x_d$), causing an internal pressure that strains the particle. Our calculation show that the strain energy associated with such a deformation is superior to that associated with the decay. Hence no decay, pressure nor volume expansion are predicted.

In short, due to the core-shell morphology, expansion into the void requires significant strain. The particle cannot exist in a state that simultaneously expands into the void and has zero strain energy. Thermodynamically comparing the energies of the decomposed, expanded, void-filling state with that of the pristine SE state, it is the latter that is more energetically favorable. Therefore the core shell structure is stable at zero pressure, with no tendency to expand. We expanded discussion of this in the revised manuscript.

(3) It is puzzling that the solid electrolytes are not tested in realistic solid state battery geometries, as this is where one would like to see the advantage. To proof the concept, solid state batteries should have been prepared and electrochemically cycled, preferably with higher voltage cathodes.

It is worth noting that our manuscript is to report an advanced sulfide solid electrolyte rather than a solid state battery, i.e., our focus is on the materials rather than the devices. Especially considering that in this very new field there is not a standard geometry of solid state battery yet, we feel that a realistic solid state battery is not a well-defined concept like the conventional liquid electrolyte coin cell. Different groups reported very different battery constructions, which were all called solid state batteries. For example, some batteries were made by a mixture of ceramic and gel electrolytes (6), some by a mixture of ceramic and liquid electrolyte/ionic

liquids (7,8), some by pure ceramic powder but of many different types (9,10) and some by a mixture of ionic salts/ceramic and polymers (11,12) . From technological point of view, it is still too early to claim which geometry is going to win the competition in this field. Our opinion is that any battery cell construction that contains the solid electrolyte component is worth trying and reporting as solid state battery, as long as it can show the performance beyond the conventional liquid cell. We just gave an example by replying review's valuable question#1, where our semi-solid-state battery shows the lithium conduction ability of our stable LSPS solid electrolyte (Figure 1.1 and Figure 1.4) that cannot be reached by any conventional liquid cell.

We would like to further point out that our semi-solid-state battery construction has certain advantages over all the previously reported geometries, especially if the research focus is on the materials level. Many solid state battery geometries suffer from the problematic interfaces between electrode and solid electrolyte layers, which can cause the artifacts to impede the study of the intrinsic properties of materials. Our battery construction on the contrary is not only without such problems, but also advantageous in many ways. Lithium metal is the perfect anode standard to locate the decomposition voltage of solid electrolyte on the cathode side. The chemical potential of Lithium remains constant during charge/discharge processes. The glassfiber well separates the lithium metal from the solid electrolyte with the liquid electrolyte inside the glassfiber mediating the lithium ion conduction. The interface on the anode side is thus trouble free, giving us the opportunity to focus on the cathode. Meanwhile, we can use the high voltage potentialstatic holding to decompose the possible small amount of liquid electrolyte in the cathode, making the measurement truly reflect the intrinsic property of solid electrolyte at the high voltage end. We thus believe that our cell construction and testing method themselves are going to be valuable to the field, in addition to the materials properties and theoretical understandings reported in our manuscript.

However, per the reviewer's request, we show here the result on all-solid-state battery using our LSPS as solid electrolyte, without any glassfiber separator layer. To achieve this goal, special treatment was performed on Li foil to form a protective layer so that the interfacial reaction between LSPS solid electrolyte and Li can be avoided. Results (Figure 1.5) show that all-solid-state battery using LTO+LSPS460+C cathode can be charged to 4V smoothly, while the all-solid-state battery using LTO+LSPS500+C cathode failed at ~3.6V due to a much more severe decomposition of LSPS500. These all-solid-state battery results correspond very well to those obtained from our semi-solid-state battery configuration, confirming again that LSPS solid electrolytes synthesized with different conditions do have different electrochemical stabilities.

Making use of the same solid-state battery configuration, solid-state CV test was also performed. Figure 1.6 shows the CV scan results in the voltage range of 1-5V for all-solid-state batteries using LTO+LSPS460+C and LTO+LSPS500+C cathode, respectively. The peaks of LTO at 1.5V exist for both batteries. However, no decomposition can be observed for LSPS460, while the decomposition of LSPS500 starting at ~3.5V is very strong and obvious. These results match with the CV results obtained by liquid cell, confirming the different electrochemical stabilities of various LSPS solid electrolytes. This is additional evidence to show that our CV test result in the original manuscript reflects the true electrochemical stability of LSPS.

More importantly, the solid-state battery test and CV results confirm that our semi-solid-state battery configuration can truly reflect the essential material property, rather than introducing artifacts. In other words, our semi-solid-state battery configuration is equivalent to the all-solid-state configuration in terms of evaluating material properties, such as the voltage stability of solid electrolytes. The new data on all-solid-state battery performance, CV results and relevant experimental details have been added to the modified manuscript.

Figure 1.5: 1st-cycle Charge/discharge profiles of all-solid-state batteries using either (a) LTO+LSPS460+C or (b) LTO+LSPS500+C as cathode, LGPS as solid electrolyte and LPS-coated Lithium foil as anode. Current rate is 0.1C for both batteries.

Figure 1.6: CV results of all-solid-state batteries using either (a) LTO+LSPS460+C or (b) LTO+LSPS500+C as cathode, LGPS as solid electrolyte and LPS-coated Lithium foil as anode. Sweeping rate is 0.1 mV/s.

The reason we choose LTO as the cathode is that LTO has no interface reaction issue in direct contact with sulfide solid electrolyte, while most high voltage cathode materials suffer from it. For example, it is known that LCO has to be coated with other types of materials to prevent such reaction. The choice of coating materials and different coating techniques will influence the battery performance, making it difficult to evaluate the intrinsic voltage stability of solid electrolyte of interest here. Also, evaluating the interface reaction stability of sulfide electrolyte is beyond the scope of the current paper. However, per the reviewer's request, we show here our result on the different performance of the coated LCO versus uncoated LCO with two different types of solid electrolyte LSPS460 (stable at high voltage) and LSPS500 (unstable at high voltage) in Figure 1.7. It shows that the coated LCO can work well with our LSPS460 (Figure 1.7b), but not LSPS500 (Figure 1.7c), while uncoated LCO (Figure 1.7a) cannot work with even the stable solid electrolyte (such as LSPS460). The example nevertheless clearly demonstrates the different high voltage stabilities of sulfide solid electrolytes, consistent with the trend we reported using LTO cathode in the original manuscript.

Figure 1.7: The charge and discharge profiles of the solid-state battery using Li metal anode and (a) uncoated bare LCO + LSPS460 + C +PTFE cathode (b) coated LCO +LSPS460+C+PTFE cathode (c) coated LCO + LSPS500+C+PTFE cathode, which can not be charged above 4.1V due to severe decomposition of LSPS500. Current rate is 0.1C.

- [6] Jae-Kwang Kim, Young Jun Lim, Hyojin Kim, Gyu-Bong Cho and Youngsik Kim A hybrid solid electrolyte for flexible solid-state sodium batteries. *Energy Environ. Sci.*, **8**, 3589 (2015).
- [7] Z. Zhang, Q. Zhang, J. Shi, Y. S. Chu, X. Yu, K. Xu, M. Ge, H. Yan, W. Li, L. Gu, Y. S. Hu, H. Li, X.Q. Yang, L. Chen, X. Huang, A Self - Forming Composite Electrolyte for Solid - State Sodium Battery with Ultralong Cycle Life, *Adv. Energy Mater.*, **7**, 1601196 (2017).
- [8] Han, X., Gong, Y., Fu, K.K., He, X., Hitz, G.T., Dai, J., Pearse, A., Liu, B., Wang, H., Rubloff, G., et al.. Negating interfacial impedance in garnet-based solid-state Li metal batteries. *Nat. Mater.* **16**, 572–579, (2016).
- [9] Xiayin Yao, Deng Liu, Chunsheng Wang, Peng Long, Gang Peng, Yong-Sheng Hu, Hong Li, Liquan Chen, and Xiaoxiong Xu, High-Energy All-Solid-State Lithium Batteries with Ultralong Cycle Life, *Nano Lett.*, **16** (11) 7148–7154 (2016).
- [10] Fudong Han, Jie Yue, Cheng Chen, Ning Zhao, Xiulin Fan, Zhaohui Ma, Tao Gao, Fei Wang, Xiangxin Guo, and Chunsheng Wang, Interphase Engineering Enabled All-Ceramic Lithium Battery, *Joule* **2**, 1–12, (2018).
- [11] Qiang Ma, Juanjuan Liu, Xingguo Qi, Xiaohui Rong, Yuanjun Shao, Wenfang Feng, Jin Nie, Yong-Sheng Hu, Hong Li, Xuejie Huang, Liquan Chen and Zhibin Zhou, A new Na[(FSO₂)(n-C₄F₉SO₂)N]-based polymer electrolyte for solid-state sodium batteries, *J. Mater. Chem. A*, **5**, 7738-7743, (2017).
- [12] Li, Y., Xu, B., Xu, H., Duan, H., Lu, X., Xin, S., Zhou, W., Xue, L., Fu, G., Manthiram, A., and Goodenough, J.B. Hybrid polymer/ garnet electrolyte with a small interfacial resistance for lithium-ion batteries. *Angew. Chem. Int. Ed.* **56**, 753–756 (2017).

(4) Above 3 V, but most likely already below, part of the Li will be extracted (the solid electrolyte will be oxidised), which should be expected to influence the calculated pressures and stability window significantly, however this is not considered.

The calculated stability windows are based on the lithium grand canonical phase diagram. As such, both Li extraction (oxidation) and insertion (reduction) are considered for all voltages between 0-5V. At 3.0V, for example, the predicted oxidation reaction for $Li_{10}SiP_2S_{12}$ is:

The calculated decomposition energy for this reaction is -0.521 eV/atom, implying a significant instability if in a non-core-shell morphology. The reaction strain however is 0.215 , leading to a reaction strain energy of 0.659 eV/atom. Since this reaction strain energy is larger than the decomposition energy, we predict that the core-shell morphology will inhibit this oxidation reaction. We add a brief statement in the corresponding paragraph to make this point clearer.

(5) In the “Characterization of core-shell microstructure” section, what is the exact composition of the core and shell materials. It appears that the core-shell structures are obtained by direct mechanical milling process. Typically for these type of materials the pure phase is obtained by the high rotation speed milling process. These ‘pure’ phase is a core-shell structure, where the core is typically more crystalline and the shell more amorphous. This may also have a strong influence on the apparent stability, which may be of large relevance for this work. More analysis of the core shell structure therefore appears necessary.

The compositions of the core and shell for different LSPS solid electrolytes were already presented and discussed in Figure 2 of our original manuscript, characterized by STEM-EDX. The atomic composition between Si, P and S in the core agrees with the nominal formula composition, considering the limit of EDX standardless technique. The compositions in the core are also close among different LSPS samples. The shell compositions, however, are different with either Si rich (e.g. LSPS460, synthesized at 460 °C with high voltage stability) or Si deficient (e.g. LSPS500, synthesized at 500 °C with high voltage instability). Note that we only control the nominal composition of the whole LSPS core-shell particle by the ratio of precursors in synthesis. Although we won’t say what’s measured by EDX is the exact composition of either core or shell based on the principle and practice of EDX, the EDX resolution is enough to disclose the Si composition trend in the shell among different LSPS samples, as we discussed in Figure 2 of the original manuscript. We also pointed out in our revised manuscript that the shell is amorphous. We attach here the electron diffraction pattern (Figure 1.8, also added to the SI of revised manuscript), from which we obtained such statement. The XRD Bragg diffraction signals are thus from the crystalline core, which show close lattice parameters among different LSPS samples, consistent with the close composition in the core measured by our STEM-EDX.

Figure 1.8: TEM diffraction pattern on the shell of LSPS solid electrolyte, showing the amorphous structure.

Discussion of the reaction kinetics for the formation of such core-shell structure during the high temperature synthesis is not the major point of our manuscript. It is possible that the ball milling procedure also contributed to the core-shell formation as the reviewer mentioned. However, we want to emphasize and clarify the originality of our core shell work here. Although core-shell or glass-ceramic structure was reported before in this field, there wasn’t such correlation between the element composition (Si or Ge in this case) in the shell and the electrochemical voltage stability of the core-shell sulfide electrolyte. There also lacks a theoretical model that explains

how such increase in Si composition in the shell can increase the voltage stability of solid electrolyte.

The key idea here is the rigidity of the shell inhibits the lithium reaction and decomposition via mechanical stabilization, as we described in the original manuscript. We also proved here that the electrolyte can be stabilized by such mechanism at high voltages. This is different from the chemical stabilization by the passivation layer idea, where a self-limiting decay or reaction on the surface of the solid electrolyte forms an inertial layer to protect the interior of the particle. This exact idea of passivation layer won't work, because such layer actually dis-functionalizes the solid electrolyte in the core by impeding the lithium conduction.

In order to be useful in a lithium conductive system, any passivation layer must itself be lithium conductive. Accordingly, any viable passivation layer must not alter the lithium chemical potential within the bulk of the solid electrolyte particle. Given that the lithium stability window of bulk solid electrolyte, as defined by a lithium grand canonical ensemble, is determined solely by the lithium chemical potential (and temperature/volume), the adoption of a lithium potential conserving passivation layer will have no impact on the chemical tendency of the solid electrolyte to decay. In fact, such passivation layer can only inhibit the reactions with elements for which it cannot conduct.

However, if the passivation layer is made to be rigid, in which case our core-shell morphology is recovered, then the passivation layer can inhibit lithium reactions via mechanical, rather than chemical, stabilization. We added a brief discussion about this in the revised manuscript.

(6) Recent literature has shown the electrochemical activity of sulphide electrolytes, which may explain why the discharge capacity of solid-state battery using the LSPS-Cl 460 electrolyte delivers a larger discharge capacity than the theoretic capacity of LTO (165mAh/g) in Figure 4b. It is therefore vital that the materials are analyzed after charge, and after cycling to proof true larger electrochemical stability.

We thank the reviewer's comments. We performed the following experiments to show that the minor-decomposition materials have true larger electrochemical stability:

- a. We disassembled the batteries after CV tests between 1V and 5V and took the photos of the separators. The color of separator remains almost unchanged (Figure 1.9b) for minor-decomposition material (such as LSPS460), while for severe-decomposition material (such as LSPS500) the separator becomes black (Figure 1.9c). Compositional analyses on the black region will be discussed later.

Figure 1.9: Pictures of (a) Original separator; (b) Separator in the battery for minor-decomposition material (LSPS460) after CV test between 1V and 5V. (c) Separator in the battery for severe-decomposition material (LSPS500) after CV test between 1V and 5V. (d) CV test result for minor-decomposition material (LSPS460) between 1V and 5V. (e) CV test result for severe-decomposition material (LSPS500) between 1V and 5V.

- b. We performed the CV scan of another battery in a narrower voltage range between 1 and 3.5 V to avoid the decomposition current peak between 3.5V and 5V for severe-decomposition material (LSPS500). This time, the separator remains to be white (Figure 1.10b)

Figure 1.10: (a) CV test result for severe-decomposition material (LSPS500) between 1V and 3.5V to avoid the high voltage decomposition. (b) Picture of separator in the battery for severe-decomposition material (LSPS500) after CV test between 1V and 3.5V.

- c. To explore the true electrochemical stability of our stable/minor-decomposition material (LSPS460) at high voltages, we expanded the CV test window from 1-5V to 1-10V, where the integrated current density is 4.91562 VA/g (Figure 1.11a). The value is still much smaller than that of severe-decomposition material (LSPS500) scanned within a much narrower window of 1 and 5V (187.98299 VA/g). The corresponding separator after the CV scan between 1 and 10V (Figure 1.11b) remains almost white with some very small black regions, indicating a small-degree of decomposition of LSPS.

Figure 1.11: (a) CV test result for minor-decomposition material (LSPS460) between 1V and 10V. (b) Picture of separator in the battery for minor-decomposition material (LSPS460) after CV test between 1V and 10V.

- d. To further confirm that the black color on separator in Figure 1.9c is due to decomposition of LSPS, we performed SEM EDS analyses on the black region of the separator (Figure 1.12), which shows the existence of Si, P, S and Cl with their composition summarized in Table 1.1. We can easily conclude that at least S should be from the decomposition of LSPS.

Figure 1.12: SEM image of the black region on the separator in Figure 1.9c and the corresponding energy dispersive spectrum.

Table 1.1: Quantitative SEM EDS analysis of the spectrum in Figure 1.12 for all the detected elements from the black region on the separator of Figure 1.9c.

Element	Wt%	At%
CK	49.98	65.63
OK	13.27	13.08
FK	08.94	07.42
SiK	01.82	01.02
PK	04.62	02.35
SK	21.00	10.33

All the data shown in the reply to this particular question are added to the SI.

Based on the fundamental issues raised I conclude that, although the concept sounds interesting, I do not consider it proven by the reported experiments and much more experiments need to be conducted.

We understand the reviewer's concern and, as discussed above, have added more experimental evidence to prove our concepts or even exceed our computational predictions.

Reviewer #2 (Remarks to the Author):

The authors have performed a combined experimental and theoretical study exploring the possibility of using compression from a 'shell' on a sulphide Li-ion solid-electrolyte 'core' to suppress reductive decomposition of the solid-electrolyte, thereby increasing its electrochemical stability window. I think the idea proposed is very interesting. As such I think it is warranted for publication in Nature Materials.

We appreciate the reviewer's positive comments.

I have a few comments in general, noted below, as well as concerns about the computations presented in this manuscript, especially the description of the computational work, which I feel needs to be addressed before the paper is accepted:

1. It is not clear if the plot shown in SI Fig. 11 & 12 is obtained by computing $H(p)$ (Eq. 3) for just LGPS OR for LGPS AND all the decomposition products? This is important to know so that one can ascertain the claims made in the paper.

For isobaric calculations (SI fig 11 & 12), $H(p)$ is calculated at every pressure for both LGPS and decomposition products. Then the convex hull is determined using eq 3.

In isovolumetric calculations, we are interested in the onset of decay, which is equivalent to there being zero radial pressure ($\lim x_d \rightarrow 0, p = 0$). We have added this detail to the revised manuscript.

2. There is also not much details about how the constant pressure DFT calculations were performed. Citing Ref. 25 is not adequate to reproduce these results. In fact, I suggest the authors explicitly report the $H(p)$ vs p graphs for all the phases considered.

Constant pressure DFT calculations were performed by applying an isotropic stress to the material during both relaxations and scf calculations (PSTRESS tag). By doing so, the prescribed pressure is added to the stress tensor during relaxation and the work pV is added to the output energy. We list the table here and also in the revised SI of our manuscript, as suggested by the reviewer (a graph is a bit difficult to read with these many compounds.) We also have added

some more details in the description of the computational method in the corresponding paragraph.

System	H(0GPa)/eV	H(1GPa)/eV	H(10GPa)/eV	H(20GPa)/eV
Ge	-4.487796E+00	-4.338350E+00	-3.079492E+00	-1.803821E+00
GeS	-9.148459E+00	-8.882685E+00	-6.765160E+00	-4.691484E+00
GeS ₂	-1.369779E+01	-1.317950E+01	-9.331516E+00	-6.359788E+00
Li ₂ S	-1.196982E+01	-1.168203E+01	-9.323823E+00	-7.015825E+00
Li ₃ P	-1.391181E+01	-1.354894E+01	-1.057828E+01	-7.679334E+00
Li ₃ PS ₄	-3.509377E+01	-3.406774E+01	-2.613501E+01	-1.958275E+01
Li ₄ GeS ₄	-3.818960E+01	-3.713570E+01	-2.861505E+01	-2.036525E+01
Li ₄ P ₂ S ₆	-5.372166E+01	-5.240342E+01	-4.165673E+01	-3.110949E+01
Li ₁₅ Ge ₄	-5.202579E+01	-5.012603E+01	-3.529954E+01	-2.133391E+01
LGPS	-1.080921E+02	-1.050711E+02	-8.132142E+01	-5.876544E+01
P	-5.378355E+00	-5.252024E+00	-4.295360E+00	-3.397391E+00
S	-4.061113E+00	-3.889834E+00	-2.767311E+00	-1.745577E+00
P ₂ S ₅	-3.260306E+01	-3.135725E+01	-2.295823E+01	-1.531866E+01
LiP ₇	-4.097063E+01	-3.990250E+01	-3.163096E+01	-2.382346E+01
Li ₃ P ₇	-4.721150E+01	-4.599375E+01	-3.635586E+01	-2.774939E+01
LiP	-8.365624E+00	-8.172500E+00	-6.580108E+00	-5.014365E+00
LiGe	-6.956550E+00	-6.756500E+00	-5.100235E+00	-3.458691E+00
Li ₉ Ge ₄	-3.962443E+01	-3.836025E+01	-2.812853E+01	-1.824771E+01

3. The mobility of li-ions should also depend on the pressure, and this will in turn influence the conductivity, in addition to the stability. In fact pressure should reduce mobility. Can the authors incorporate possible change in kinetics to clearly discuss the pros- and cons- of using such a core-shell structure to improve chemical stability?

We agree with the reviewer that pressure will influence the conductivity. This can be seen in light of *Energy & Environmental Science* **6**, 148-156, (2013) in the reference of our original manuscript, where a 2% change in the lattice parameter reduced the conductivity of LGPS by six

orders of magnitude. However, as discussed above to reply to review#1's question#2, and also in our original manuscript, the core-shell morphology is not pressurized. Only after decomposition will the particle experience pressure. If the applied battery voltage is within the stability window of the core-shell structure, the infinitesimal pressure associated with the tendency to decompose will prevent the decomposition from happening. We revised the corresponding paragraph in the resubmitted manuscript to make this point clearer.

4. Much of the known superionic Li-ion solid-electrolytes have intrinsic site-occupancy disorder, which is known to influence the high mobility seen in these materials (e.g. J. Mater. Chem. A, 2017,5, 1153-1159). Can the authors comment on how this site occupancy could possibly be affected by the induced pressure from the shell ? (this is somewhat related to comment-3 above).

This is a very intriguing question. In light of the question 3, we are not generally worried about the effect of pressure on site occupancy as it pertains to conductivity. This is because we do not expect the pressurization to happen at all within the predicted stability windows.

However, this does bring to light another effect of the core-shell morphology that could lead to additional stability beyond the strain energy. Pressurization is anticipated to lead to lithium ordering, and hence reduced lithium configurational entropy and ionic conductivity. Therefore, the pressurization increases the energy of the particle not only by straining the neighborhood, but also by ordering. If this is accounted for, the free energy requirement for the local stability of solid electrolyte becomes:

$$\delta\Delta G_{decomposition} = \frac{\partial\Delta G_{chem}}{\partial x_d} \delta x_d + \frac{\partial W}{\partial x_d} \delta x_d - \frac{\partial T\Delta S_{conf}}{\partial p} \frac{\partial p}{\partial x_d} \delta x_d > 0$$

Given that lowering the entropy (ordering the lithium sites) will always require an increase in energy, we see that this term will add to the work term expanding the range in which the chemical reaction energy is not dominant. This effect together with some other effects we qualitatively discussed in the revised manuscript could explain why we see even greater stability in the oxidation region than was predicted purely based on the core-shell constant volume argument. However, it is worth noting that the core-shell effect is still the major effect to stabilize the solid electrolyte, in the sense that without which all other effects won't have the opportunity to apply. We have added such discussion in the revised manuscript.

Reviewer #3 (Remarks to the Author):

This paper describes the importance of amorphous phase in LGPS superionic conducting solid electrolyte. The properties of sulfide solid electrolytes can be controlled and improved by modifying the microstructures. The amorphous shell covering the LSPS core improves the stability window of the composite solid electrolyte particles. It seems that description of originality is excessive. Glass ceramic solid electrolytes, which contain high ion conducting crystals in the glass matrix, have been studied the most. The authors should state progress and new ideas for prior research with proper references. Li₂S-P₂S₅ glass-ceramic solid electrolytes with LGPS phase has already been reported around 2002 although the crystal structure of the

LGPS phase has been clarified since 2011. Since there are many other concerns, I do not recommend publication in the present stage.

We respect the reviewer's knowledge on the history of the field. In the revised manuscript, we discussed the major progress, concepts and results for prior researches in sulfide solid electrolytes. The three categories of sulfide solid electrolytes, i.e. sulfide glasses, sulfide glass-ceramics and sulfide crystals are concisely discussed and properly cited in the introduction part of the modified manuscript.

Although the structure of the crystalline LGPS or LSPS has been studied by diffraction techniques, change in the synthesis details from various groups may decorate the same crystalline phase with different microstructures of amorphous or glassy phases. This is very possible considering that the sulfide system with mixed glass-ceramic phases can exist from synthesis, while there lacks a systematic check and understanding about the effect of such amorphous/glassy phases on the materials properties in the previous reports. An understanding of how these possible microstructures can control the electrochemical properties of crystalline sulfide electrolytes, especially the voltage stability and interface compatibility, is hence critical.

The goal of our paper is to demonstrate that the control and modification of the microstructures in LGPS and LSPS can adjust and improve their voltage stability. More importantly, we aim to reveal the underlying mechanism between the microstructure and performance of sulfide solid electrolyte, which can serve as the guidelines for the future design. We want to clarify that we did not claim the originality of our work on the core-shell or glass-ceramic structure. We claimed the originality of our core-shell structure with Si rich shell can largely improve the voltage stability (as in the reply to reviewer#1's questions), while Si-deficient or S-rich shell cannot. We are the first to understand and report the major mechanism of this new phenomenon by our theoretical model based on the combined DFT and thermodynamic simulations. We try to make this point clearer in the revised manuscript that the originality won't be misunderstood.

1: The major claims are unclear in this manuscript.

Our major claim is that the electrochemical high voltage stability of our sulfide solid electrolyte can be largely improved via modified microstructures. We proved this experimentally and understood the mechanism by DFT-based modeling work.

2: The evidence of decomposition in oxidation should be shown.

We thank reviewer's comment, and this is the same question raised by reviewer#1's question#6. We have added the evidence to the SI of the revised manuscript.

3: What is "glass fiber" inserted between Li and solid electrolyte? If the glass fiber includes liquid electrolytes, the impregnation with electrolyte is most concerned.

The reviewer understand it correctly that there was a glassfiber layer to separate the lithium metal from reacting with the sulfide solid electrolyte, and a small amount of liquid electrolyte was dropped to the glassfiber, as we described in the Method of the original manuscript. We

argue the value of such battery construction in reply to reviewer#1's question#3, where the major argument is that such construction is ideal for the study of the intrinsic properties of solid electrolyte on the cathode side, which minimizes the artifact from the interface issues in many other solid state battery geometries. As to the concern about the impregnation raised here, it is very relevant to reviewer#1's question#1, where we used the potentialstatic high voltage holding at 3V for 24 hours to decompose the specially-designed liquid electrolyte within the cathode, leaving the reported results truly reflecting the property of solid electrolyte. Results (Figure 1.3) show the same trend for minor- and severe-decomposition LSPS, despite the fact that liquid electrolyte in the cathode has been decomposed. Furthermore, we constructed all-solid-state battery using LSPS as solid electrolyte without the glassfiber separator layer or liquid electrolyte, to prove again that various LSPS solid electrolytes have different electrochemical stabilities (Figure 1.5-1.6). These results confirm that our semi-solid-state battery configuration reported in the original manuscript can truly reflect the essential material property, rather than introducing artifacts. In other words, our semi-solid-state battery configuration is equivalent to the all-solid-state configuration in terms of evaluating material properties, such as the voltage stability of solid electrolytes.

4: Details on DFT calculations are not provided.

We have provided detailed original theoretical formalism in both the main text and the SI, which is the foundation for the implementation of our DFT calculation. We provided some necessary DFT details to understand our work, while we referred the routine DFT details to the Materials Project. However, we understand the reviewer's concern that such organization may not be the best to the general readers. We thus added some additional details in both the reply here and the revised manuscript.

Constant pressure calculations of enthalpy were performed by calculating the energy of each crystal phase with an externally applied isotropic pressure during both relaxation and scf calculations. This was performed at each pressure reported (0, 1, 10, 20 GPa). The Gibbs energy of each phase was then calculated at each chemical potential of interest in post-processing.

$$G(p, \mu) = H(p) - \mu_{Li}N_{Li}$$

For isobaric conditions, where the core-shell morphology is expected to operate at near zero pressure ($1 \text{ atm} \approx 0 \text{ GPa}$), the relevant free energy was constructed in post-processing:

$$\Phi(\mu) = G(0, \mu) = E - \mu_{Li}N_{Li}$$

In each case, convex hull methods were used to find the predicted chemical decomposition energy and the resulting products. For constant pressure, this energy was reported directly (SI Figure 15 & 16). For core-shell, the chemical decomposition energy was combined with the mechanical work in post-processing to determine the total decomposition energy as reported in Figure 6 in the original manuscript.

5: Since this paper deals with concrete discussions, interest in researchers except battery community is considered to be relatively small.

The theory part of this paper talks about how the voltage stability of a material is improved by the mechanical stabilization mechanism under the lithium grand canonical ensemble, which may have broader impact in materials science beyond the battery community. The experimental proof of the high voltage stability of our sulfide solid electrolyte will generate the enthusiasm in the battery field. Last but not least, the semi and pure solid-state cell configurations/ testing method themselves are going to be valuable to the field, in addition to the materials properties and theoretical understandings reported in our manuscript.

6: Some references should be added and discussed.

We added proper references and discussions as suggested by reviewer. We added introductions on the development history of sulfide solid electrolytes, and discussed the typical characteristics for each three category of sulfide solid electrolytes, including sulfide glasses, sulfide glass-ceramics and sulfide crystals.

7: It seems that description of the originality is excessive. An intelligible explanation is important, but originality should be described accurately.

We understand the reviewer's concern and we clarified the originality of our work in the reply to your general comment. We also modified the manuscript accordingly to avoid the misunderstanding on this point.

8: There are many unclear points in experiments and results.

We tried to improve the clarity based on the reviewers' comments in the revised manuscripts.

Reviewers' comments:

Reviewer #1 (Remarks to the Author):

The authors have provided significant additional evidence for the presented results, and these in the revised version. This takes away my concerns and makes the manuscript in my view very interesting and ready for publication.

Reviewer #2 (Remarks to the Author):

The authors have addressed my major concerns satisfactorily, and have made the appropriate changes in the revised manuscript. I feel that the current manuscript is acceptable for publication.

Reviewer #3 (Remarks to the Author):

The manuscript has been revised well in accordance with the reviewers' comments. However, some concerns which should be addressed still remain.

1.

The stability of the shell materials is more important than that of core material.

2.

The theoretical study on the change of the resistance to decomposition under applied pressure is interesting and reasonable.

On the other hand, there is no evidence for controlling swelling of the particles using a core-shell structure as the reviewer#1 of the original manuscript commented.

3. The shell materials should not be rigid enough although the reviewer partly agrees with the idea that the decomposition compress the remaining pristine materials. Local stresses of at least 1 GPa or more must be applied continuously.

About the description "Given the high modulus of amorphous Si (biaxial modulus ~140 GPa), it is likely that such a high Si:P ratio would lead to both a core of significantly decreased compressibility and a very rigid shell.":

The elastic modulus should not change dramatically.

4. The reviewer appreciates that the authors showed the results of the all-solid-state batteries without any glass fiber separator.

However, the contents on special treatment should be shown some what more clearly, if the authors can, to mitigate concerns of the reviewers #1 and #3.

5. The authours commented that they did not claim the originality of thier work on the core-shell or glass-ceramic structure.

The reviewer think that there is a discrepancy between the title and abstract and the contents. The title is "Advanced Sulfide Solid Electrolyte by Core-Shell Structural Design", but the importance of the core-shell structure has not been revealed yet.

Point-to-Point Response To Reviewers' Comments

Reviewers' comments:

Reviewer #1 (Remarks to the Author):

The authors have provided significant additional evidence for the presented results, and these in the revised version. This takes away my concerns and makes the manuscript in my view very interesting and ready for publication.

We thank the reviewer's suggestion for publication.

Reviewer #2 (Remarks to the Author):

The authors have addressed my major concerns satisfactorily, and have made the appropriate changes in the revised manuscript. I feel that the current manuscript is acceptable for publication.

We thank the reviewer's suggestion for publication.

Reviewer #3 (Remarks to the Author):

The manuscript has been revised well in accordance with the reviewers' comments. However, some concerns which should be addressed still remain.

We thank the reviewer's positive comments that our manuscript has been revised well in accordance with the reviewers' comments. We further address your additional concerns as following:

1. The stability of the shell materials is more important than that of core material.

The reviewer understands our work correctly. The stability of the shell is very important as we demonstrated in this work. It was shown that the voltage stability of crystalline sulfide-ceramic solid electrolytes can be greatly improved by the implementation of a rigid shell that restricts the expansion of solid electrolyte particles. It follows that if the shell is not stable in a battery configuration, then the shell will decompose and, hence, not be able to stabilize the solid electrolyte core. Since we have shown in experiment that silicon rich amorphous shell structures are of adequate stabilities to lead to significant opening of the stability window, the stability of the shell materials has been demonstrated.

2. The theoretical study on the change of the resistance to decomposition under applied pressure is interesting and reasonable.

On the other hand, there is no evidence for controlling swelling of the particles using a core-shell structure as the reviewer#1 of the original manuscript commented.

We thank the reviewer's comment that our theory is interesting and reasonable. Based on this theory, swelling of the particles will only occur when decay or decomposition happens. In other

words, swelling is equivalent to decomposition of sulfide solid electrolyte particles. As discussed in our manuscript and previous rebuttal letter, we have provided combined theoretical and experimental evidences to show that our core-shell structure can inhibit decomposition or swelling of particles, and such capability depends on the composition of the shell.

Theoretically, we have calculated the thermodynamics of solid electrolyte decay from an unstrained solid electrolyte state ($p = 0$) to a strained (compressed, $p > 0$) state. While the solid electrolyte particle could expand into the surrounding void, it could only do this by decaying, causing an internal pressure that strains the particle. Our calculations show that within the stability window the strain energy associated with such a deformation is superior to that associated with the decay. Hence no decay, no pressure nor particle swelling are predicted within the stability window.

Experimentally, we showed the severe decomposition (swelling of particles) of less stable materials with weaker shell (such as LSPS500), as manifested by black separator (SI Figure 11c) after CV test. In contrast, stable materials with strong shell (such as LSPS460) shows no decomposition (swelling of particles), since the color of separator remains almost unchanged (SI Figure 11b). The black color on separator was proved to be caused by decomposition (swelling of particles) of sulfide solid electrolyte, based on quantitative EDX analyses on the black region in SI Figure 11c, as shown in SI Figure 14. Furthermore, if the decomposition (swelling of particles) of the weak material (such as LSPS500) is minimized (by performing CV test in a narrower voltage range between 1 and 3.5 V, such that decomposition current peak between 3.5V and 5V is avoided), the separator remains to be white (SI Figure 12b). Therefore, the above experimental evidence shows that the decomposition (swelling of particles) can be prevented by core-shell structure with appropriate composition.

SI Figure 11: Pictures of (a) Original separator; (b) Separator in the battery for minor-decomposition material (LSPS460) after CV test between 1V and 5V. (c) Separator in the battery for severe-decomposition material (LSPS500) after CV test between 1V and 5V.

SI Figure 12: (a) CV test result for severe-decomposition material (LSPS500) between 1V and 3.5V to avoid the high voltage decomposition. (b) Picture of separator in the battery for severe-decomposition material (LSPS500) after CV test between 1V and 3.5V.

3. The shell materials should not be rigid enough although the reviewer partly agrees with the idea that the decomposition compress the remaining pristine materials. Local stresses of at least 1 GPa or more must be applied continuously.

About the description "Given the high modulus of amorphous Si (biaxial modulus ~140 GPa), it is likely that such a high Si:P ratio would lead to both a core of significantly decreased compressibility and a very rigid shell.":

The elastic modulus should not change dramatically.

I. We would like to note that the reviewer's comment that "local stresses of at least 1GPa or more must be applied continuously" is not necessarily true. This argument follows the same lines as the authors' response to reviewer #1's question about "expansion into the void" in our previous response letter, which takes away his concern.

A potential misunderstanding of this theory is that the solid electrolyte, when held at a given voltage (V'), decays/pressurizes until the resulting pressure is sufficient as to open the stability window to include V' . For example, considering LGPS (SI Figure 15), it may be thought that should the solid electrolyte be held at 1.25V at an external pressure of 0GPa, then the particle would decompose until the local pressure reaches 10GPa at which point 1.25V enters the stability window and decomposition halts. Based on this approach, local stress of >1GPa would be needed to see any meaningful improvements in stability.

However, this approach mischaracterizes the effects of a [quasi]constant volume ensemble. SI Figure 15 shows the energy of the constant pressure reaction, which is distinctly different than the case of a [quasi]constant volume reaction of interest (as presented in Fig 6), where the pressure increases upon decomposition by Δp . With constant pressure the change in volume at a fixed pressure is what opens the window, while with [quasi]constant volume it is the combination of volume expansion AND an increase in the pressure that stabilizes the window.

The net result is that, for the [quasi]constant-volume cases given in Figure 6, an infinitesimal change in decomposition fraction (δx_d) requires an infinitesimal increase in strain energy (δU_{strain}) such that $\delta U_{strain} \geq |\delta \Delta G_{reaction}|$. Accordingly, zero decomposition/pressurization occurs at zero ambient pressure within the predicted voltage window in Figure 6. This discussion has been added to the revised manuscript.

II. In order to address reviewer #3's additional concern regarding the extent of shell rigidity needed for noticeable stability improvements, we made some analysis here. A spherical core-shell model with the morphology and composition consistent with our TEM observation corresponds to a β_{shell} value of around $2 \times 10^{-2} \text{ GPa}^{-1}$, giving an estimated voltage stability window of 1.7 – 3.1 V based on Figure 6b. The estimation here used a shell modulus of around 75GPa, which is obtained based on an unusually high Si composition of around 30 at% to 40 at% as characterized by our STEM-EDX (Figure 2) by using supplementary Figure 17. In comparison the Si composition in the core of our crystalline LSPS is only less than 10 at%. It should also be noted that even if the shell modulus is comparable to the core (20-30 GPa), the stability window of a core-shell-structured solid electrolyte is still significantly larger than that without shell at an estimated β_{shell} value of around 10^{-1} GPa^{-1} based on Figure 6b. We thank reviewer's comment and have added this paragraph to the revised manuscript.

4. The reviewer appreciates that the authors showed the results of the all-solid-state batteries without any glass fiber separator.

However, the contents on special treatment should be shown some what more clearly, if the authors can, to mitigate concerns of the reviewers #1 and #3.

We described the details for all-solid-state battery assembling in our previous revised manuscript (Methods part, highlighted by yellow color). We copy it here now:

For all-solid-state batteries, lithium anode was coated with a protection layer in the following steps: Firstly, a solution was prepared by mixing Li_2S_6 (0.5M) and P_2S_5 (1M) in the mixture solvent of DME and DOL (1:1, v/v) in the glovebox for 1h. Then lithium foil (5/16 inch in diameter) was soaked into the prepared solution for 2h. The soaked lithium foils were assembled into Li/Li symmetric battery for 5 cycles of charge-discharge, with 1M LiTFSi, 0.025M Li_2S_6 , and 0.05M P_2S_5 in DME and DOL (1:1, v/v). Finally, the lithium foils after the above process were taken out of the symmetric battery for use as the anode (hereafter LPS-Li). The cathode of the all-solid-state cell is a mixture of LTO+LSPS+carbon black powder (75:20:5), and solid electrolyte is pure LGPS powder. 0.5mg of cathode mixture powder and 80mg of solid electrolyte powder were pressed together to form a disc-shape pellet, which is pressed onto a freshly prepared LPS-Li in a Swagelock Cell to form an all-solid-state battery. The same battery configuration was used for solid-state CV test with 0.1mV/s sweeping rate.

5. The authours commented that they did not claim the originality of thier work on the core-shell or glass-ceramic structure.

The reviewer think that there is a discrepancy between the title and abstract and the contents. The title is "Advanced Sulfide Solid Electrolyte by Core-Shell Structural Design", but the importance of the core-shell structure has not been revealed yet.

The authors believe that the novelty of this paper stems from the experimental identification and the theoretical understanding of volume constriction as the method of stabilization in core-shell structures. We have identified a new method to decrease the decomposition energy (by adding a strain factor $\Delta G_{rxn} \rightarrow \Delta G_{rxn} + \Delta U_{strain}$) and provided a clear mechanism for the design of such shells that can lead to the improvements in the stability window. We have revealed and discussed in depth for the importance of the core-shell structure in stabilizing crystalline sulfide solid electrolyte, which can serve as guidelines for future solid electrolyte design. Based on the above explanations and response to reviewers' comments, we authors think, to the best of our understanding, that the current title is appropriate to the content of the manuscript.

REVIEWERS' COMMENTS:

Reviewer #3 (Remarks to the Author):

Thank you for the authors' response.

Reviewer #4 (Remarks to the Author):

In view of this reviewer, the authors satisfactorily addressed the concerns raised upon previous review, in this way improving the quality of the manuscript, while convincingly showing that volume constriction as a method of stabilization in core-shell structures can be beneficial for achieving better electrochemical properties. In agreement with previous reviewer 1, the authors have provided evidence for their presented results, thus rendering the revised manuscript suitable for publication.

Reviewers' comments:

Reviewer #3 (Remarks to the Author):

Thank you for the authors' response.

Thank you for reviewing the paper.

Reviewer #4 (Remarks to the Author):

In view of this reviewer, the authors satisfactorily addressed the concerns raised upon previous review, in this way improving the quality of the manuscript, while convincingly showing that volume constriction as a method of stabilization in core-shell structures can be beneficial for achieving better electrochemical properties. In agreement with previous reviewer 1, the authors have provided evidence for their presented results, thus rendering the revised manuscript suitable for publication.

We thank the reviewer's suggestion for publication.